# A SWI/SNF-dependent transcriptional regulation mediated by POU2AF2/C11orf53 at enhancer

Aileen Szczepanski[1,2,5], Natsumi Tsuboyama[1,2,5], Huijue Lyu[1,2,5], Ping Wang[1,2], Oguzhan Beytullahoglu[1,2], Te Zhang[1,2], Benjamin David Singer [1,2,3], Feng Yue [1,2,4], Zibo Zhao [1,2] ✉ & Lu Wang [1,2] ✉

Recent studies have identified a previously uncharacterized protein C11orf53 (now named POU2AF2/OCA-T1), which functions as a robust co-activator of POU2F3, the master transcription factor which is critical for both normal and neoplastic tuft cell identity and viability. Here, we demonstrate that POU2AF2 dictates opposing transcriptional regulation at distal enhance elements. Loss of POU2AF2 leads to an inhibition of active enhancer nearby genes, such as tuft cell identity genes, and a derepression of Polycomb-dependent poised enhancer nearby genes, which are critical for cell viability and differentiation. Mechanistically, depletion of POU2AF2 results in a global redistribution of the chromatin occupancy of the SWI/SNF complex, leading to a significant 3D genome structure change and a subsequent transcriptional reprogramming. Our genome-wide CRISPR screen further demonstrates that POU2AF2 depletion or SWI/SNF inhibition leads to a PTEN-dependent cell growth defect, highlighting a potential role of POU2AF2-SWI/SNF axis in small cell lung cancer (SCLC) pathogenesis. Additionally, pharmacological inhibition of SWI/SNF phenocopies POU2AF2 depletion in terms of gene expression alteration and cell viability decrease in SCLC-P subtype cells. Therefore, impeding POU2AF2-mediated transcriptional regulation represents a potential therapeutic approach for human SCLC therapy.

Lung cancer remains a prominent contributor to global cancer morbidity and mortality worldwide[1], with small-cell lung carcinoma (SCLC) representing a particularly aggressive and lethal subtype[2–4]. SCLC is characterized by its development within lung tissues and is frequently diagnosed at advanced stages with metastatic spread to other organs, accounting for roughly 13% of all lung cancer cases[5]. Regrettably, the patient survival rate for SCLC is among the lowest of all cancer subtypes, and the prognosis for patients afflicted with this condition is often bleak.

Human SCLC has been identified as a highly malignant tumor of primitive neuroendocrine (NE) cells with inactive mutations in Rb1 and Trp53 tumor suppressors[6–8]. However, recent studies on molecular subtyping further classified human SCLC based on the relative expression of four key lineage-specific transcription and co-

[1]Department of Biochemistry and Molecular Genetics, Feinberg School of Medicine, Northwestern University, Chicago, IL 60611, USA. [2]Simpson Querrey Center for Epigenetics, Feinberg School of Medicine, Northwestern University, Chicago, IL 60611, USA. [3]Division of Pulmonary and Critical Care Medicine, Department of Medicine, Feinberg School of Medicine, Northwestern University, Chicago, IL 60611, USA. [4]Robert H. Lurie Comprehensive Cancer Center, Feinberg School of Medicine, Northwestern University, Chicago, IL, USA. [5]These authors contributed equally: Aileen Szczepanski, Natsumi Tsuboyama, Huijue Lyu. ✉e-mail: zibo.zhao@northwestern.edu; lu.wang1@northwestern.edu

transcription regulators: achaete-scute homolog 1 (ASCL1; SCLC-A subtype), neurogenic differentiation factor 1 (NEUROD1; SCLC-N subtype), yes-associated protein 1 (YAP1; SCLC-Y subtype), and POU class 2 homeobox 3 (POU2F3; SCLC-P subtype)[9]. Notably, the SCLC-Y subtype cells appeared to be enriched for intact RB by immunohistochemistry. Therefore, whether high YAP1 represents a transcriptional driver of this subtype, or a subtype-specific correlation has yet to be determined[10].

Distinguishing between subtypes of lung cancer is of great importance, as the effectiveness of treatment could vary substantially among these subtypes. Recently, we and other colleagues utilized a global genome-wide CRISPR screening database (DepMap) and have identified and characterized human SCLC subtype-specific essential factors and signaling pathways within all SCLC subtypes. Intriguingly, we have identified *C11orf53* gene, which is the top essential factor for SCLC-P subtype cells[11–13]. We then worked with HUGO gene nomenclature committee to rename this gene *POU2AF2* (POU Class 2 Homeobox Associating Factor 2), based on its robust function as a co-activator of the master transcriptional regulator POU2F3[14]. By utilizing multiple unbiased genome-wide studies, we discovered that more than 90% of POU2AF2 occupies distal enhancer elements and regulates the tuft cell-specific gene signature at their nearby super-enhancers and maintains the cell identity[11–14]. However, why loss of POU2AF2 triggers cell growth inhibition and cell death remains to be discovered.

In our current studies, we uncover an unexpected role of POU2AF2 in maintaining cell viability by regulating PTEN expression, and further elucidate the mechanism of how POU2AF2 interacts with the SWI/SNF chromatin remodeling complex at H3K27me3 enriched distal enhancer elements and facilitates the transcriptional repression function of Polycomb repressive complex 2 (PRC2).

## Results

### POU2AF2 elicits opposing effects of gene expression at distinct distal enhancer elements

Previously, we have characterized the chromatin occupancy of POU2AF2, which co-localizes with POU2F3 at the chromatin (Fig. 1a, Supplementary Fig. 1a) and functions as its co-activator across the genome[11]. Despite being a nuclear protein lacking a known canonical chromatin binding domain, POU2AF2 is shown to be recruited by the POU domain of POU2F3[11]. In our ChIP-seq analysis, we have identified 6,987 overlapping POU2F3 and POU2AF2 peaks in the NCI-H526 SCLC cell line. Notably, approximately 93% of these peaks occupy enhancers that are characterized by high levels of H3K4me1 (Fig. 1a, Cluster 2 and 3). Interestingly, we have observed the subdivision of these enhancers into two clusters through k-means clustering: Cluster 2 peaks were enriched with the active enhancer histone mark H3K27ac, while Cluster 3 peaks were more enriched with the poised enhancer histone mark H3K27me3. This phenomenon was also observed in a different SCLC cell line, NCI-H211 (Supplementary Fig. 1b). To determine how POU2AF2 and POU2F3 regulate gene expression at distinct enhancer clusters, we first compared the gene expression profile between wild-type and POU2F3 or POU2AF2-depleted cells. The scatter plots showed a significant correlation between POU2F3 and POU2AF2 target genes in different cell lines (Fig. 1b, Supplementary Fig. 1c, d). It has been shown that depletion of either POU2AF2 or POU2F3 leads to the repression of genes involved in cell identity such as tuft-cell lineage genes[12,14]. Therefore, we further conducted pathway analysis with the up-regulated genes in POU2F3/POU2AF2-depleted cells. Interestingly, we found that most of these genes were enriched in cell differentiation or development pathways (Fig. 1c), implying there is a distinct role of POU2F3/POU2AF2 in transcriptional activation or repression. Next, we integrated the RNA-seq results with the ChIP-seq data from Fig. 1a and analyzed the expression change of genes nearest to POU2F3 and POU2AF2 overlapped peaks at the genome-wide level. As expected, loss of either POU2AF2 or POU2F3 reduced the expression of these genes (e.g., tuft cell-specific genes) nearest to Cluster 2 peaks, which were enriched with active enhancer histone marks H3K4me1 and H3K27ac (Fig. 1d, left and middle panel). This phenomenon is consistent with the effects of JQ1 treatment (Fig. 1d, right panel, Fig. 1e).

Surprisingly, we found that the up-regulated genes in POU2AF2 (with median RPM 13.78 (sg*NONT*), 15.42 (sg*POU2AF2*−1), and 16.03 (sg*POU2AF2*−2) or POU2F3 depleted cells were dramatically enriched at Cluster 3 peaks that were marked by repressive histone marks H3K27me3 (Fig. 1d, Supplementary Fig. 1e). Intriguingly, these genes were not upregulated in JQ1 treated cells, indicating that the activation of Cluster 3 nearby genes may not be the secondary effect of the repression of Cluster 2 nearby genes (Fig. 1d). To study the function of POU2F3 and POU2AF2 at enhancer regions marked by H3K27me3 and determine whether there is a direct binding of POU2F3/POU2AF2 at H3K27me3-marked enhancer chromatin, we conducted motif analysis and identified 35,503 enhancer regions marked by H3K4me1 and H3K27ac (Fig. 1f, left panel), and 6,538 enhancers marked by H3K4me1 and H3K27me3 in NCI-H526 cells (Fig. 1f, right panel). Consistent with our previous studies, POU2F3 DNA binding motif (5′-TATGCAAATC-3′)[11,15] is the most enriched motif in both regions (Fig. 1f). Similar results were observed at specific gene loci (Fig. 1g, Supplementary Fig. 1f) in different SCLC cell lines. These results revealed a potential bidirectional transcriptional regulatory function of POU2F3/POU2AF2 at different enhancers.

### POU2AF2 is essential for PRC2 maintenance and repression of Polycomb target genes

Previous studies have demonstrated that a subclass of enhancers enriched for H3K4me1 (but lacking H3K27ac) is also marked by H3K27me3 and bound by PRC2 in human or mouse pluripotent cells. These elements, referred to as "poised enhancers", are located near key developmental genes[16–20]. Given the presence of H3K27me3 at enhancer regions (Cluster 3), we postulated that POU2AF2 may be directly or indirectly involved in transcriptional repression at these sites through a potential collaboration with EZH2, the dominant enzymatic catalytic subunit of PRC2 responsible for catalyzing H3K27me3 histone modification.

To investigate this hypothesis, we treated cells with EZH2-specific inhibitor GSK126 and observed a similar effect on transcriptional activation that phenocopies POU2AF2 depletion (Fig. 2a), without affecting the protein levels of POU2AF2 (Supplementary Fig. 2a). In particular, these genes nearest to the Cluster 3 peaks were strongly upregulated, compared to genes nearest to Clusters 1 and 2 peaks (Fig. 2b, c). The co-upregulated genes by POU2AF2 depletion or GSK126 treatment were also enriched in differentiation and developmental pathways (Fig. 2d, Supplementary Fig. 2b), suggesting a potential co-function between POU2AF2 and the PRC2 complex in human small cell lung cancer cells.

We further evaluated the molecular mechanisms underlying the loss of POU2AF2 and its impact on Polycomb target genes. We found no obvious reduction in total EZH2 or SUZ12 protein levels in POU2AF2-depleted cells (Supplementary Fig. 2c). However, the ChIP-seq assay revealed a significant reduction in H3K27me3 levels at Cluster 3 regions, but not at Cluster 1 or 2 loci (Fig. 2e, Supplementary Fig. 2d), indicating that POU2AF2 depletion may attenuate the occupancy or catalytic activity of PRC2 at these sites.

Next, we conducted ChIP-seq to determine the chromatin occupancy of EZH2 and SUZ12 levels in cells with or without POU2AF2. The results indicated a dramatic reduction in chromatin occupancy of EZH2 and SUZ12 at Cluster 3 peaks in POU2AF2-depleted cells (Fig. 2f, g, Supplementary Fig. 2e–g), consistent with the reduction in

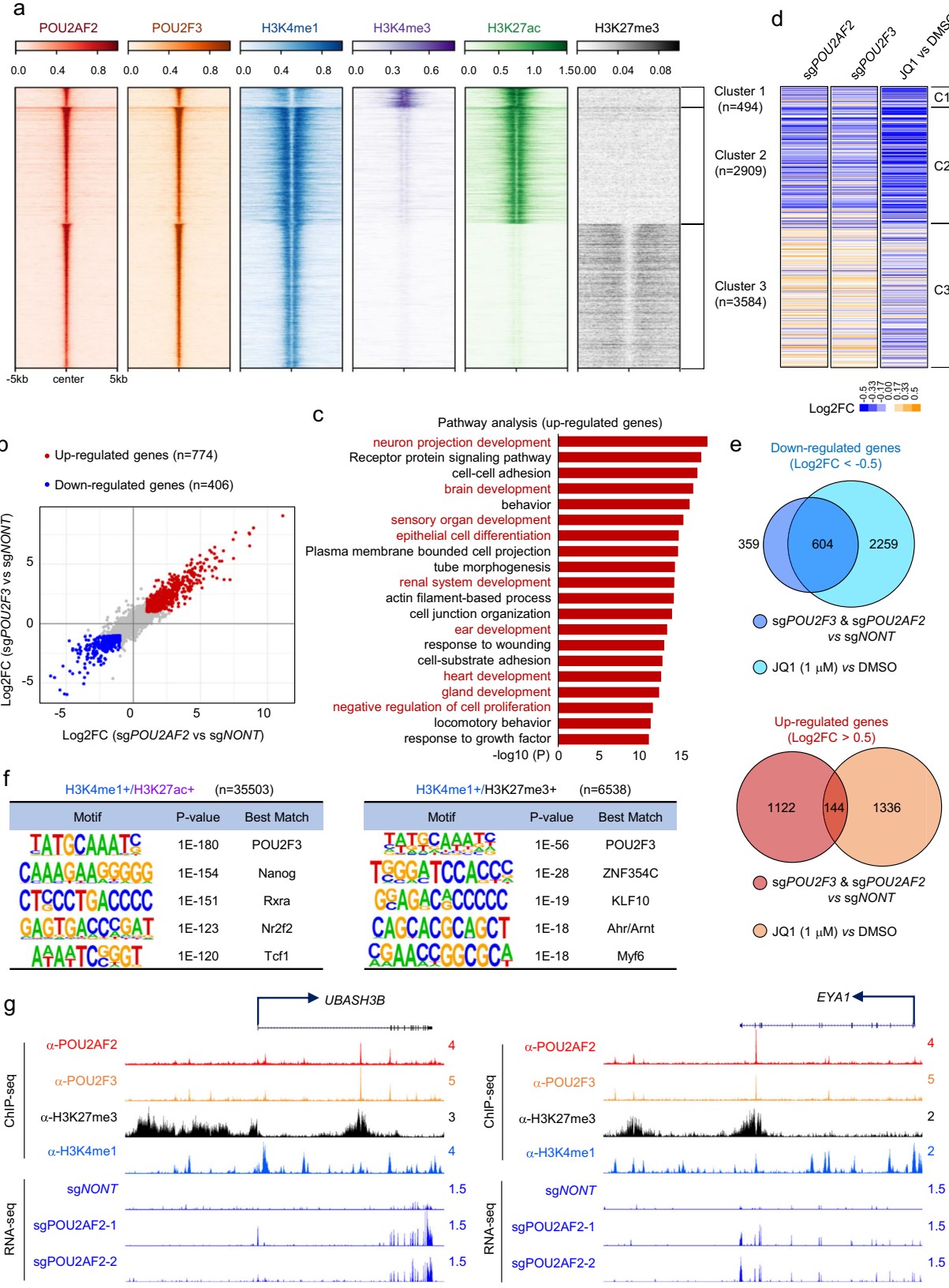

**Figure (panels a–g)**

H3K27me3 levels observed (Fig. 2e). Notably, the significantly reduced H3K27me3 (Fig. 2h) and EZH2 peaks (Fig. 2i) tend to be broader than the other peaks, indicating that POU2AF2 (narrow peaks) may not physically interact with the PRC2 complex or directly recruit EZH2 to the chromatin. Finally, we did not detect a significant increase of H3K27ac levels at the Cluster 3 peaks after POU2AF2 depletion

(Supplementary Fig. 2h), despite elevated gene expression, which is consistent with recent studies demonstrating that enhancer H3K27ac levels may not be directly associated with the nearby gene expression[21]. The activation of the poised enhancer in cancer cells may be different from that in embryonic stem cells and may not require the deposition of H3K27ac[18].

**Fig. 1 | POU2AF2 elicits opposing effects of gene expression at distal enhancer elements. a** The POU2AF2 and POU2F3 ChIP-seq was conducted in human SCLC cell line NCI-H526 cells. The overlapping POU2AF2 and POU2F3 peaks ($n$ = 6987) were divided into three clusters by k-means clustering based on POU2AF2, POU2F3, and histone marks (H3K4me1, H3K27ac, H3K4me3, and H3K27me3). The ChIP-seq signal of these histone marks were further centered on the three clusters. **b** The scatter plot shows the correlation of gene expression change when POU2F3 or POU2AF2 were depleted by sgRNAs. The significantly altered genes (|log2FC|>1, adj. $p < 0.01$) by POU2F3/POU2AF2 depletion were highlighted in red (upregulated, $n$ = 774) or blue (downregulated, $n$ = 406). Data are derived from two biological replicates. Genes with Benjamini-Hochburg adjusted $p$-values less than 0.01 were considered to be differentially expressed in the EdgeR analysis[48]. **c** Metascape pathway enrichment analysis with up-regulated genes in both POU2F3 and POU2AF2 depleted cells. The -log10(P) value was calculated by Metascape software v3.5[49]. **d** The heatmap shows the nearby gene expression change aligned to the corresponding ChIP-seq peak (Fig. 1a) in either POU2AF2 depleted, POU2F3 depleted, or JQ1 treated NCI-H526 cells. Data are derived from two biological replicates. **e** The Venn diagram shows the overlap between JQ1 target genes and POU2F3/POU2AF2 common target genes. **f** The enhancer regions marked by H3K4me1 and H3K27me3 ($n$ = 6538), or H3K4me1 and H3K27ac ($n$ = 35503) were identified in NCI-H526 cells. The motif analysis shows the most significant motifs enriched in each type of enhancer elements. HOMER screens its library of reliable motifs against the target and background sequences for enrichment, returning motifs enriched with a p-value less than 0.05[55]. **g** The track examples show the occupancy of POU2F3 and POU2AF2 at H3K4me1 and H3K27me3 marked enhancer regions, and the activation of nearby gene expression upon the loss of POU2AF2.

## POU2AF2 interacts with the SWI/SNF complex and regulates chromatin accessibility

To further elucidate the mechanism by which POU2AF2 mediates the recruitment of EZH2 to chromatin and whether there is a direct interaction between POU2AF2 and PRC2, we employed GFP-tagged POU2AF2 purification followed by mass spectrometry analysis in both SCLC NCI-H526 and non-SCLC HEK293T cell lines. Our analysis yielded 44 enriched proteins common to both cell lines (Fig. 3a) but we did not detect any subunits of the Polycomb complexes. However, we found multiple subunits of the SWI/SNF complex to be significantly enriched in the POU2AF2 co-purified proteins from both cell lines (Supplementary Fig. 3a). Immuno-precipitation experiments with the POU2AF2-specific antibody confirmed a strong interaction between POU2AF2 and different subunits of the SWI/SNF complex in NCI-H526 cells (Fig. 3b). Consistent with the mass spectrometry results, we did not observe a detectable interaction between POU2AF2 and EZH2, despite the genetic interaction observed between both factors in Fig. 2. To further study the composition of endogenous POU2AF2 and the SWI/SNF complex, nuclear extracts from NCI-H526 cells were subjected to size exclusion (SE) chromatography, followed by western blot analysis of the elution profiles of POU2AF2 and SWI/SNF complex. As shown in Supplementary Fig. 3b, we observed that a large fraction of POU2AF2 was co-eluted with SWI/SNF components. These results led us to hypothesize that POU2AF2 may impact chromatin accessibility via its interaction with the SWI/SNF complex. Following confirmation that protein levels of major subunits within the SWI/SNF complex were not affected after POU2AF2 depletion (Supplementary Fig. 3c), we conducted ChIP-seq with a BRG1-specific antibody in POU2AF2 wild-type and depleted cells to investigate how the loss of POU2AF2 regulates the function of the SWI/SNF complex at the chromatin level. Our analysis revealed a drastic redirection of chromatin-bound BRG1 after POU2AF2 depletion, with a gain of 6,649 and a loss of 21,957 peaks of BRG1 across the whole genome in POU2AF2-depleted cells (Fig. 3c, Supplementary Fig. 3d). Interestingly, the motif analysis demonstrated that both the retained and lost peaks of BRG1 were significantly enriched with the POU2F3 motif, which was not observed in the gained BRG1 peaks (Fig. 3d).

Recent studies have provided mechanistic and conceptual connections between the SWI/SNF chromatin remodeling complex and genome structure and organization[22–25]. Therefore, to investigate how higher order chromatin structure may influence gene expression change induced after POU2AF2 depletion, we performed Hi-C in SCLC cells transduced with either non-targeting sgRNA or two independent POU2AF2 specific sgRNAs. The majority of A and B genomic compartment occupancy was unaltered upon POU2AF2 depletion, but regions with differential compartmentalization primarily undergo compaction in POU2AF2-depleted cells versus control cells or "shift" from the more open A to the more closed B chromatin compartment: 3.9% of bins "shifted" from A to B and 2.7% from B to A (Fig. 3e). To correlate A/B compartment switching with gene expression, we examined the change in expression of genes located within the B-to-A or A-to-B bins. A significant downregulation in POU2AF2-depleted cells relative to control was observed for genes located within A-to-B bins, whereas gene expression was slightly increased in B-to-A bins (Fig. 3f, Supplementary Fig. 3e). The APA plots also revealed more sgNONT specific loops than sgPOU2AF2 specific loops (Fig. 3g). These results suggested that the attenuation of the POU2AF2-SWI/SNF axis changed the higher order chromatin structure, adding another layer of gene expression alteration that potentiates the epigenetic alteration.

To further understand the direct relationship between POU2AF2 and BRG1, we compared the BRG1 chromatin occupancy between POU2AF2 wild-type and depleted cells by centering on the POU2F3 and POU2AF2 overlapped peaks as defined in Fig. 1a. Notably, we found that BRG1 co-localized with POU2AF2 across the genome, and depletion of POU2AF2 by two distinct sgRNAs significantly reduced the chromatin occupancy of the vast majority of BRG1 peaks (Fig. 3h; Supplementary Fig. 3f-h). As shown in Fig. 3i, there is a strong reduction of BRG1 peaks, which co-localized with POU2F3, POU2AF2, H3K27me3, and H3K4me1 peaks at Cluster 3-gene locus, and loss of POU2AF2 leads to a reduction and redistribution of BRG1. Consistently, the ATAC-seq results showed a reduction of chromatin accessibility in both Cluster 2 and Cluster 3 (but less in Cluster 1) regions, which may be attributed to the loss of BRG1 occupancy in those regions (Fig. 3j).

## SWI/SNF inhibition phenocopies POU2AF2 depletion on histone modifications and gene expression

In mouse embryonic stem cells (mESCs), enhancer elements may exist in distinct epigenetic states, including active, primed, or poised[20]. Poised enhancers are characterized by open chromatin and marked by two opposing histone modifications, namely, the active mark H3K4me1 and the repressive mark H3K27me3[19,20]. Our previous investigations have shown that similar enhancer elements in human SCLC cells are marked by different histone modifications[26].

To determine whether the ATPase activity of SWI/SNF is required for maintaining enhancer states in SCLC cells, we treated NCI-H526 cells with BRM014, a widely used small-molecule inhibitor that specifically targets BRG1/BRM ATPase activity of the SWI/SNF complex[27]. As shown by the RNA-seq analysis, we found that there is a significant positive correlation between POU2AF2 and SWI/SNF inhibitor target genes in the same cell line (Fig. 4a). The data showed that inhibiting the SWI/SNF complex led to the repression of genes located near H3K4me1/H3K27ac positive enhancer elements (Cluster 2) and the activation of genes near H3K4me1/H3K27me3 positive enhancer elements (Cluster 3) (Fig. 4b). This gene expression pattern is similar to that observed in POU2AF2-depleted cells (Fig. 1d). A total of 430 genes were

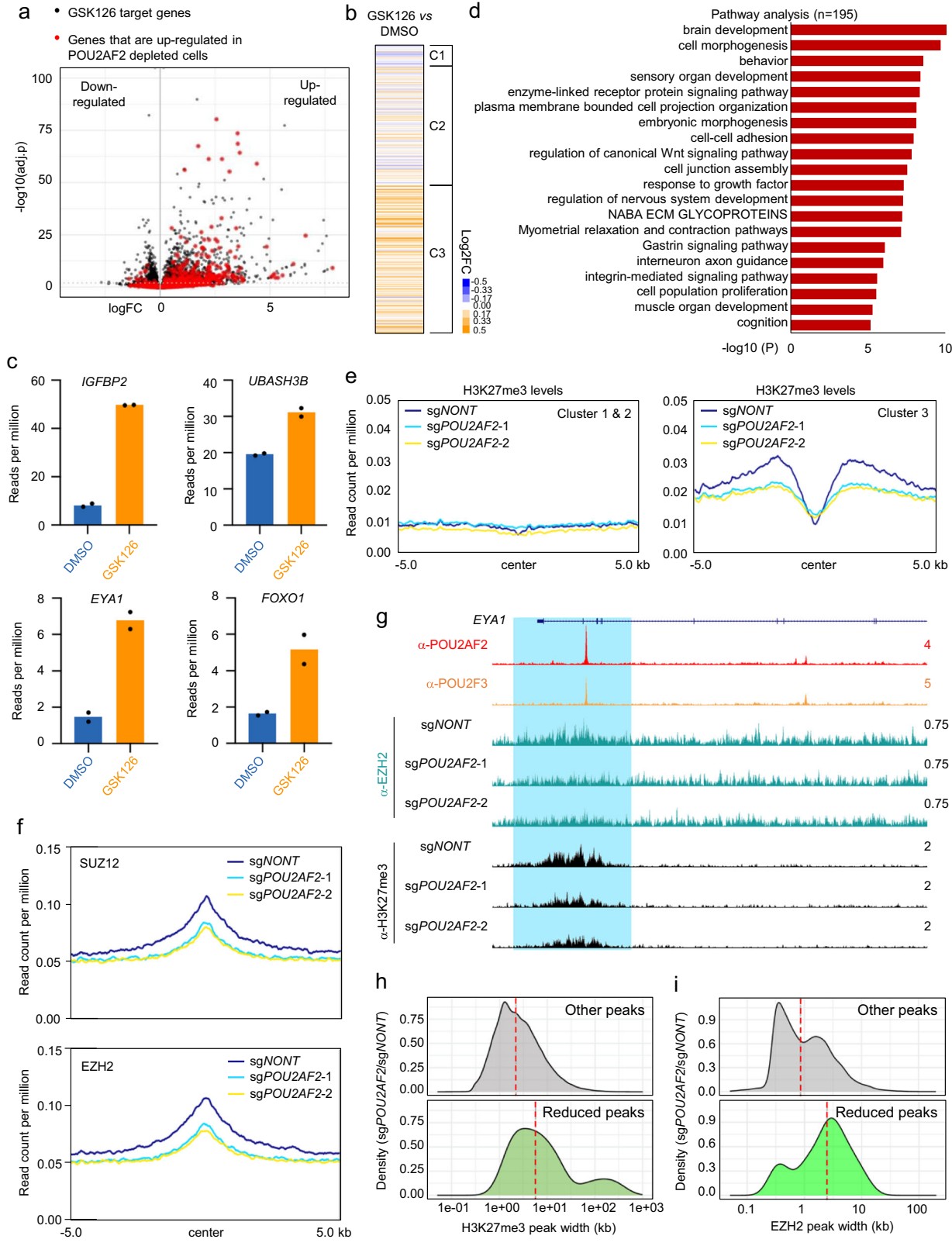

down-regulated in both BRM014-treated and POU2AF2-depleted cells, while 419 genes were up-regulated in both BRM014 treated and POU2AF2-depleted cells (Supplementary Fig. 4a). These changes primarily involve pathways related to differentiation and development (Supplementary Fig. 4b, c).

Our ATAC-seq analysis of BRM014-treated cells also revealed a decrease in open chromatin accessibility (Fig. 4c) that resembled

observations previously seen in Clusters 2 and 3 of POU2AF2-depleted cells (Fig. 3h). Interestingly, inhibiting the catalytic activity of the SWI/SNF complex resulted in a remarkable reduction of active enhancer marks H3K4me1/H3K27ac at Cluster 2 peaks (Fig. 4d) leading to significant transcriptional repression (Fig. 4e, f, Supplementary Fig. 4d), which phenocopies POU2AF2 depletion (Supplementary Fig. 4e, f).

**Fig. 2 | POU2AF2 is essential for PRC2 maintenance and repression of Polycomb target genes. a** NCI-H526 cells were treated with EZH2 inhibitor GSK126 (2 μM) for 6 days. The volcano plot shows the overlap between EZH2 inhibitor GSK126 target genes and the up-regulated genes (log2FC > 1) in POU2AF2 depleted cells as highlighted in red. Data are derived from two biological replicates. Genes with Benjamini-Hochburg adjusted p-values less than 0.01 were considered to be differentially expressed in the EdgeR analysis[48]. **b** The log2FC heatmap shows the nearby gene expression change with EZH2 inhibitor GSK126 treatment aligned to the three cluster ChIP-seq peaks defined in Fig. 1a in NCI-H526 cells. Data are derived from two biological replicates. **c** The RNA-seq data for several POU2AF2 repressed genes that were also up-regulated by GSK126 treatment. Data are derived from two biological replicates. **d** Pathway analysis with the co-upregulated genes in POU2AF2 depleted and GSK126 treated cells. The -log10(P) value was calculated by Metascape software v3.5[49]. **e** Average plots show the H3K27me3 levels at cluster 1&2 (left) and cluster 3 (right) in NCI-H526 cells transduced with either non-targeting sgRNA or two distinct POU2AF2 sgRNAs. **f** The average plots show the SUZ12 and EZH2 levels at cluster 3 peaks in NCI-H526 cells transduced with either non-targeting CRISPR sgRNA or two distinct POU2AF2 sgRNAs. **g** The track example shows that loss of POU2AF2 reduced the chromatin occupancy of EZH2 and H3K27me3 levels at the *EYA1* gene locus. Differential peak analysis was performed with EdgeR analysis. The density plots show the distribution of peak width of H3K27me3 (**h**) and EZH2 (**i**) when POU2AF2 was depleted. The green color represents the peaks that were downregulated in POU2AF2 depleted cells (*n* = 105 for H3K27me3; *n* = 112 for EZH2), and the gray color represents the rest of the peaks, including the upregulated and stable peaks (*n* = 61444 for H3K27me3; *n* = 8243 for EZH2).

Intriguingly, similar to POU2AF2 depletion, BRG1/BRM inhibition reduced EZH2 chromatin binding, and H3K27me3 levels at the Cluster 3 peaks (Fig. 4g, h, Supplementary Fig. 4g). Consequently, the decreased H3K27me3 levels led to an increase of the nearby-gene expression of Cluster 3 (Fig. 4i, Supplementary Fig. 4h). Notably, there was a slight increase of H3K27me3 levels at Cluster 1 and 2, indicating a possible Polycomb reprogramming induced by a global loss of SWI/SNF activity (Supplementary Fig. 4i).

### Loss of POU2AF2 results in cell viability decrease by activating PTEN expression

Initially, POU2F3 was identified as the master transcription factor that defines the P-subtype of human SCLC[14]. Recently, our laboratory group and others have identified POU2AF2 (originally named C11orf53) as an important co-activator of POU2F3, which maintains cell identity by transcribing a number of tuft cell-specific gene expressions at active enhancers[11–14]. In our current studies, we discovered an opposing effect of POU2AF2 at distinct enhancer elements. Pathway analysis revealed that multiple cell growth and differentiation pathways were significantly enriched with the activated genes in POU2AF2-depleted cells (Supplementary Fig. 4c). To understand the biological function of these genes activated by POU2AF2 depletion, we conducted a global dropout CRISPR screening to identify depletion of which genes could rescue the cellular growth inhibition induced by two distinct *POU2AF2*-specific sgRNAs (Fig. 5a). As a result, we identified 28 factors containing more than three guide RNAs that were enriched (Fig. 5b). We further analyzed the dependency score of each of these 28 factors in all SCLC-P cell lines to determine which factor had the strongest tumor suppressor function in SCLC-P cells (Fig. 5c). We found that *PTEN* (one of the 28 factors) was the strongest tumor suppressor gene in SCLC-P subtype cells (Fig. 5d), which was significantly activated in POU2AF2-depleted cells (Fig. 5e, f). Notably, depletion of *PTEN* by two distinct sgRNAs significantly rescued the cell death induced by POU2AF2 depletion, which is consistent with our CRISPR screening result (Fig. 5g, h). Interestingly, *PTEN* was one of the Cluster 3 genes occupied by POU2AF2 at its enhancer. Loss of POU2AF2 dramatically reduced BRG1 occupancy at the specific enhancer element enriched by H3K27me3 (Fig. 5i). These results suggest that POU2AF2/PTEN axis could be a potential therapeutic target for SCLC-P subtype.

### Inhibition of the ATPase activity of the SWI/SNF complex as a potential therapeutic strategy for SCLC treatment

The SWI/SNF complex is a multi-subunit chromatin-remodeling complex that has been implicated in cancer development due to aberrant expression and mutation of its subunits in various tumor types[28–31]. Consequently, the SWI/SNF complex may function as either a tumor suppressor or oncogene in a context-dependent, tumor-specific

manner[32]. Previous studies have demonstrated that *BRG1* depletion by shRNA is lethal in MAX-deficient human SCLC cells[33]. However, comprehensive genome wide characterization of the SWI/SNF complex in SCLC cells is necessary to better understand the complexity of its role in SCLC cells. Here, using an unbiased, whole genome-wide CRISPR screening approach, we have demonstrated that SCLC-P subtype cells are sensitive to depletion of most SWI/SNF subunits (Fig. 6a). Thus, we aimed to determine the potential anti-SCLC effect of the well characterized small-molecule inhibitor of the SWI/SNF complex. To elucidate how the ATPase activity of BRG1/BRM affects PTEN expression and cell viability in SCLC-P subtype cells, we treated three different cell lines (NCI-H526, NCI-H211, and CORL311) with BRM014. Our results showed that inhibition of BRG1 and BRM significantly reduced cell growth and viability in vitro (Supplementary Fig. 5a), with a significant increase of *PTEN* mRNA and protein levels (Fig. 6b, c). The upregulation of *PTEN* expression upon BRM014 treatment was further confirmed by RNA-seq analysis in all three different cell lines (Supplementary Fig. 5b–e).

Next, to investigate the involvement of *PTEN* in BRM014-mediated cell growth inhibition, we treated wild-type and PTEN-depleted cells with different concentrations of BRM014 for 72 hours. As shown in Fig. 6d, we found that loss of PTEN significantly attenuated the inhibitory effect of BRM014 on cell viability. Subsequently, to evaluate the efficacy of BRM014 on SCLC progression in vivo, we established a mouse xenograft model with human SCLC H526 cells. Treatment with BRM014 resulted in a significant delay in disease progression and a reduction in tumor size (Fig. 6e, f). Moreover, we confirmed the on-target effect of BRM014 in vivo by measuring the expression levels of *PTEN* and the expression of Tuft cell markers (*TAS1R3*, *IRAG2*, *AVIL*, and *NREP*) in tumor tissues (Fig. 6g), which were found to be consistent with our in vitro findings (Fig. 4e). Overall, our study highlighted a potential therapeutic strategy involving the targeting of the ATPase activity of the SWI/SNF complex in SCLC treatment (Fig. 6h).

## Discussion

Initially, we and our colleagues identified POU2AF2, which functions as a co-activator of POU2F3, a master transcriptional regulator in SCLC[12–14]. Despite lacking a classical chromatin binding domain, POU2AF2 is recruited to chromatin by POU2F3 and is essential for maintaining an open chromatin state at distal enhancer elements. Depletion of POU2AF2 significantly reduces the expression of tuft cell-specific genes by decreasing enhancer activity within SCLC-P subtype cells[12–14]. Accordingly, we have renamed this gene as the second co-activator of POU2 class transcription factors (POU Class 2 Homeobox Associating Factor 2)[11,12]. In our current studies, we have discovered an unexpected transcriptional activation effect upon POU2AF2 depletion. Specifically, loss of POU2AF2 leads to the upregulation of genes involved in the cell cycle and differentiation (Fig. 1c). Importantly, the upregulation of these genes is not a secondary effect of the cell identify

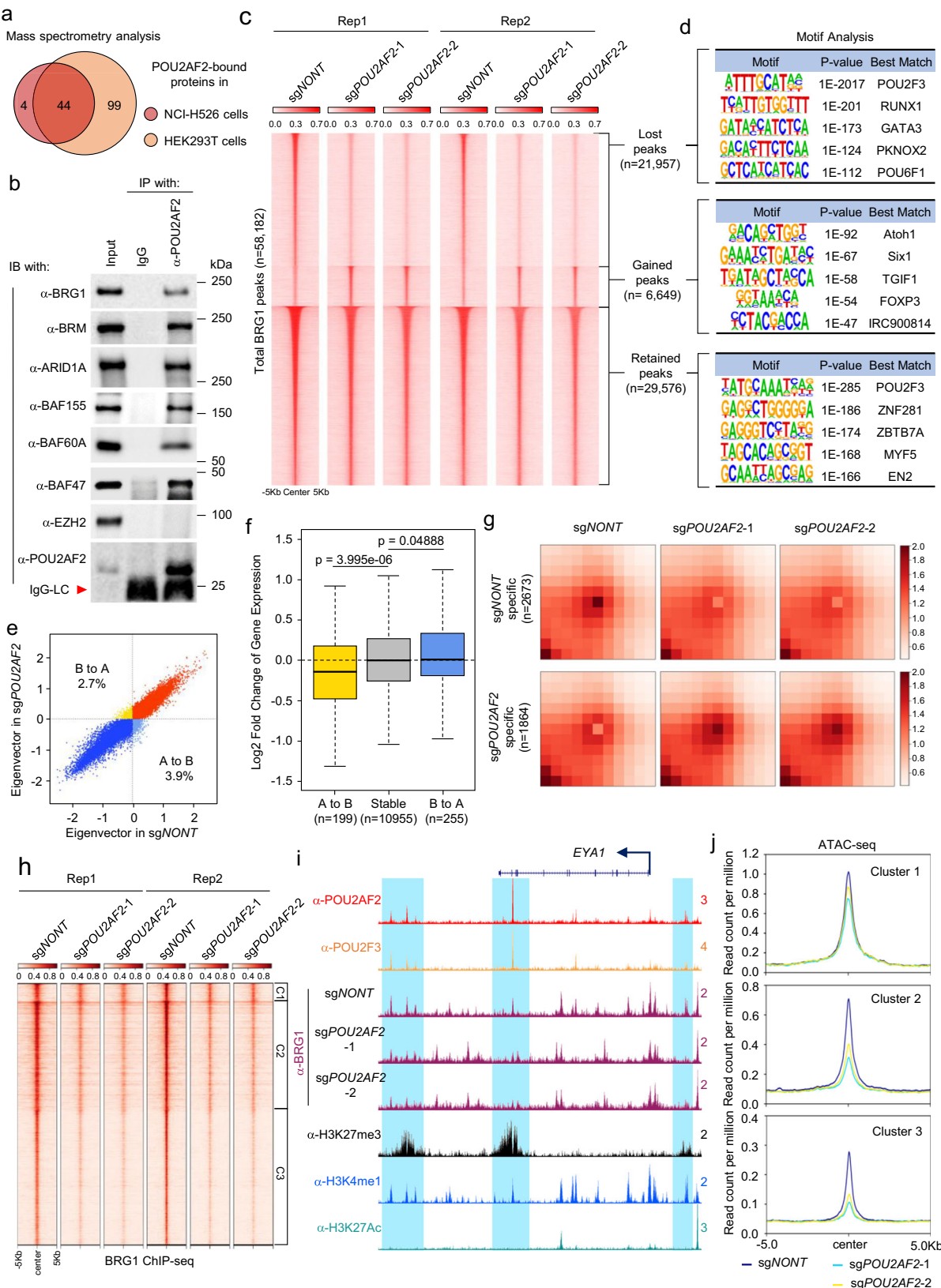

switch, as it is not observed in JQ1-treated cells, which overall display a stronger repressive phenotype throughout all three gene clusters. Therefore, in these studies, we sought to investigate the molecular mechanisms underlying the potential transcriptional repressive effect of POU2AF2 in SCLC cells and to elucidate the functional role of these genes upregulated upon POU2F3/POU2AF2 depletion.

Enhancers are crucial regulators of gene expression and can activate or repress targeted genes from a distance[34]. These elements can exist in various epigenetic states, including active, primed, or poised[17]. Poised enhancers are marked by histone H3K4me1 and H3K27me3, while primed enhancers are marked by histone H3K4me1/2, and active enhancers are marked by H3K27ac and H3K4me1[16]. Poised

**Fig. 3 | POU2AF2 interacts with the SWI/SNF complex and regulates chromatin accessibility. a** GFP-tagged POU2AF2 was purified from HEK293T and SCLC cell line NCI-H526 cells and the co-eluted protein was further subjected to mass spectrometry analysis. The Venn diagram shows the overlap of interacting proteins in both cell lines. **b** The protein-protein interaction between endogenous POU2AF2 and different subunits within the SWI/SNF complex was validated by immunoprecipitation in NCI-H526 cells. EZH2 was used as a negative control. $n = 2$ biologically independent experiments. Source data are provided as a Source Data file. **c** The NCI-H526 cells were transduced with either non-targeting sgRNA or two distinct POU2AF2 sgRNAs. BRG1 occupancy was determined by ChIP-seq with its specific antibody. The BRG1 peaks ($n = 58182$) were divided into three groups based on the differential peak analysis: peak lost in sg*POU2AF2* versus sg*NONT* ($n = 21957$); peak gained in sg*POU2AF2* versus sg*NONT* ($n = 6649$); peak retained in sg*POU2AF2* versus sg*NONT* ($n = 29576$). Data are derived from two independent biological replicates. **D** The motif analysis of the three groups with decreased BRG1 (Group 1), increased BRG1 (Group 2), and retained BRG1 (Group 3) in sg*POU2AF2* vs. sg*NONT* were performed and shown. HOMER screens its library of reliable motifs against the target and background sequences for enrichment, returning motifs enriched with a p-value less than 0.05[55]. **e** PC1 was calculated for each 100-kb genomic bin to determine its A or B compartmentalization. Compartment switching, decompaction and compaction upon POU2AF2 depletion are represented in the scatter plot. Percentage of compartment shifts (A to B or B to A) are shown. **f** Genes located in A-to-B shifted, stable and B-to-A shifted bins were selected (>= ½ gene length located within the bins). Genes with detectable expression levels were further selected (199 for A-B and 255 for B-A) for analysis of the logFC gene expression in sg*POU2AF2* versus sg*NONT*, shown in the box plot. *p*-value is calculated by two-sided Wilcoxon test. Center line: median; top and bottom hinges of box: the third and first quantiles; whiskers: quartiles ± 1.5 × interquartile range. **g** APA plot depicts aggregated signals from sg*POU2AF2*-specific loops ($n = 1864$) versus sg*NONT*-specific loops ($n = 2673$). **h** The BRG1 ChIP-seq signals in sg*NONT* or sg*POU2AF2* cells were centered on the three clusters defined in Fig. 1a. Data are derived from two independent biological replicates. **i** The track example shows the loss of POU2AF2 reduces BRG1 occupancy at the enhancer regions marked by both H3K27me3 and H3K4me1. **j** The chromatin accessibility in POU2AF2 wild-type and depleted cells was determined by ATAC-seq. The peaks were centered on the three clusters.

enhancers were defined as transcriptionally inactive elements in human and mouse embryonic stem (ES) cells[20,21]. During activation, H3K27ac is deposited to replace H3K27me3, which switches enhancer states to their active form. In human SCLC cells, we found that POU2AF2 and POU2F3 could occupy two categories of enhancer elements marked by either H3K4me1/H3K27ac or H3K4me1/H3K27me3 (Fig. 1a). Depletion of POU2AF2 or POU2F3 resulted in a significant activation of genes located proximal to H3K4me1/H3K27me3 marked enhancers. We also detected a dramatic reduction of H3K27me3 levels during gene activation, but no clear replacement of H3K27ac with H3K27me3. This finding supports recent studies indicating that H3K27ac is not always necessary for transcriptional programming[35]. Further, we found that a large number of genes repressed by POU2AF2 could also be derepressed by an EZH2 inhibitor treatment (Fig. 2a), providing evidence of a potential co-function between POU2AF2 and PRC2 at Cluster 3 loci. Indeed, depletion of POU2AF2 significantly reduced EZH2 and SUZ12 occupancy at the H3K27me3 marked enhancer regions.

Previous studies have shown conflicting and antagonistic functions of the SWI/SNF and Polycomb complexes in transcriptional regulation. For instance, loss of BAF47 (SNF5 or INI) tumor suppressor leads to elevated expression of EZH2 and strong repression of Polycomb targets[36]. In contrast, other studies have demonstrated that BRG1 represses gene expression via interaction with co-repressors, and loss of BRG1 reduces H3K27me3 levels and depresses gene expression[37]. In fact, various models have been proposed to explain how the SWI/SNF complex represses gene expression. For instance, it has been suggested that upon the rapid degradation of the SWI/SNF complex, there is a redistribution of Polycomb complexes, which further leads to a gain of chromatin accessibility and transcriptional derepression[38].

In our current study, we found that inhibiting the ATPase activity of the SWI/SNF complex in human SCLC cells led to a significant reduction in genome-wide chromatin accessibility, including both H3K27ac and H3K27me3 marked enhancer elements, similar to POU2AF2 depletion. Surprisingly, we also found that the poised enhancer-bound PRC2 complex was reduced after SWI/SNF inhibition or POU2AF2 depletion, presumably due to the reduced chromatin accessibility observed in our ATAC-seq results. It has been reported that BRG1 is required to maintain Polycomb-mediated repression of non-mesodermal developmental regulators during the mesoderm induction, suggesting cooperation between Brg1 and Polycomb complexes[39]. Our result further demonstrated that the chromatin remodeling activity of SWI/SNF is necessary for both the chromatin occupancy and the function of the PRC2 complex in human SCLC cells. Therefore,

our study suggested a molecular model of how the SWI/SNF complex is recruited by the coactivator POU2AF2 and facilitates Polycomb repression at H3K27me3 marked distal enhancer elements to repress gene expression.

It is well-established that the Class II POU domain transcription factors (POU2F1, POU2F2, and POU2F3) could both activate and repress gene expression[40–42]. For instance, in colon cancer cells, POU2F1 promotes a repressed state of gene expression through recruiting the NuRD chromatin-remodeling complex[42]. In breast cancer cells, both POU2F1 and POU2F2 could bind to the promoter of the iNOS gene, forming a higher-order complex that fails to recruit RNA polymerase[41]. Our research in SCLC-P cells demonstrated that POU2F3 and its co-activator POU2AF2 are essential for the activation of lineage-specific genes and the maintenance of tuft cell identify, and they also mediate the transcriptional repression of genes involved in development and cell cycle. The loss of either POU2F3 or POU2AF2 results in dramatic cell death and a switch in cell identity. Therefore, it is technically challenging to eliminate the possibility that the observed transcriptional activation in POU2F3/POU2AF2-depleted cells is due to cell death or a change in cell identity. Consequently, this model requires further exploration.

Dysregulation or mutations within subunits of the SWI/SNF complex has been identified as a driver in various human diseases, including cancer. Recent investigations have revealed a high prevalence of SWI/SNF-encoding gene mutations in approximately 20% of all human cancers[43,44]. Furthermore, in SWI/SNF-mutant tumors, the residual sub-complex is believed to promote oncogenic transcriptional programs[30], highlighting its potential as a therapeutic target. On the other hand, some cancer types lacking SWI/SNF mutations (e.g., small-cell lung cancer) may be susceptible to inhibition of the SWI/SNF ATPase subunit BRG1[45]. Indeed, our study has demonstrated that POU2AF2-dependent SCLC-P cell lines are highly responsive to BRG1 depletion or SWI/SNF inhibition, both in vitro and in vivo. In addition, inhibition of PRC2 activity and function by EZH2 inhibitors, which have been widely tested in clinical trials, could also activate POU2AF2-SWI/SNF axis repressed genes efficiently. Consequently, our studies have yielded insights into potential clinical interventions by therapeutically targeting the POU2AF2-mediated transcriptional programming for treating human SCLC.

## Methods
### Antibodies and Reagents
POU2F3 (#36135), H3K27ac (#8173), H3K4me1 (#5326), H3K4me3 (#9751), H3K27me3 (#9733), Histone H3 (#4499), Cleaved-PARP

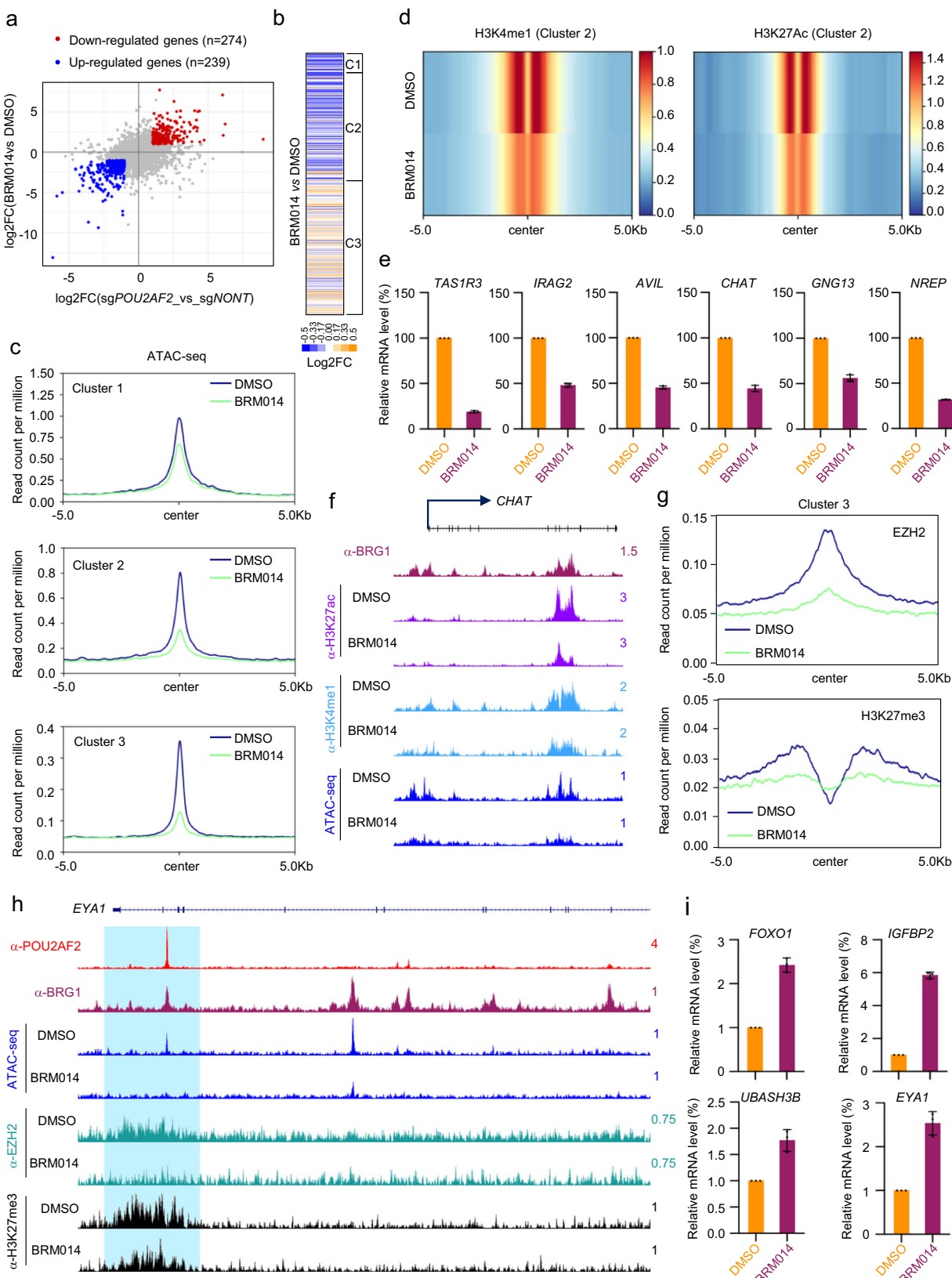

(#5625), Cleaved Caspase 3 (#9664), SUZ12 (#3737), EZH2 (#5246), BRG1 (#49360), BRM (#11966), ARID1A (#12354), BAF57 (#11956), BAF155 (#11956), BAF60A (#35070), BAF47 (#91735), and PTEN (#9188) antibodies were purchased from Cell Signaling. Tubulin antibody (E7) was purchased from the Developmental Studies Hybridoma Bank. HSP90 (sc-7947) antibody was purchased from Santa Cruz. BRG1

(ab110641) and EZH2 (ab191250) antibodies were purchased from Abcam, and were used for ChIP-seq. The POU2AF2 (C11orf53) antibody was produced in rabbits at Pocono Rabbit Farm And Laboratory Inc. by using full-length POU2AF2 recombinant protein as antigen. For all of the western blot experiments, the antibody dilution is 1: 2k. For immunoprecipitation (IP) and chromatin immunoprecipitation (ChIP),

**Fig. 4 | The ATPase activity of SWI/SNF complex is required for POU2AF2 mediated transcriptional regulation. a** The scatter plot shows the correlation of gene expression change when POU2AF2 were depleted by sgRNAs or cells were treated with BRM014. The significantly altered genes ($|$log2FC$|>1$, adj. $p < 0.01$) were highlighted in red (upregulated, $n = 239$) or blue (down-regulated, $n = 274$). Data are derived from two biological replicates. Data are derived from two biological replicates. Genes with Benjamini-Hochburg adjusted p-values less than 0.01 were considered to be differentially expressed in the EdgeR analysis[48]. **b** The log2 fold change heatmap shows the nearby gene expression change aligned to the corresponding ChIP-seq peak (Fig. 1a) in BRM014 (1 µM) treated NCI-H526 cells. Data are derived from two biological replicates. **c** The ATAC-seq experiment was conducted in NCI-H526 cells treated with either BRM014 or DMSO. The average plot shows the ATAC-seq signal centered at the three clusters. **d** The heatmap bar plot shows H3K4me1 and H3K27ac levels at Cluster 2 (active enhancer) peaks between DMSO and BRM014 treated cells. **e** The NCI-H526 cells were treated with either DMSO or BRM014 (1 µM) for 24 hours. The mRNA levels of *TAS1R3*, *IRAG2*, *AVIL*, *CHAT*, *GNG13*, and *NREP* were determined by real time PCR. $n = 3$ technical replicates. Data are presented as mean values ± standard deviation (SD). Source data are provided as a Source Data file. **f** The track example shows the H3K27ac, H3K4me1, and ATAC-seq signals in cells treated with either DMSO or BRM014 at active enhancer locus of *CHAT* gene. **g** The average plot shows the chromatin occupancy of EZH2 and H3K27me3 levels at Cluster 3 in NCI-H526 cells treated with either DMSO or BRM014. **h** The track example shows the ATAC-seq signals, H3K27me3 levels and EZH2 occupancy at *EYA1* gene locus in NCI-H526 cells treated with either DMSO or BRM014. **i** NCI-H526 cells were treated with 1 µM BRM014 for 24 hours. The mRNA levels of *FOXO1*, *IGFBP2*, *UBASH3B*, and *EYA1* were determined by real time PCR. $n = 3$ technical replicates. Data are presented as mean values ± standard deviation (SD). Source data are provided as a Source Data file.

5 µg of antibody per reaction was used. The small molecule inhibitors GSK126 (S7061) and JQ1 (S7110) were purchased from Selleck Chemicals. The small molecule inhibitor BRM014 (HY-119374) was purchased from Med Chem Express.

## Cell Lines
HEK293T cells were purchased from ATCC (CRL-3216), and maintained with DMEM (Fisher Scientific, #15013CV) containing 10% FBS (MT35010CV, Sigma). The SCLC cell lines NCI-H526 (CRL-5811) and NCI-H211 (CRL-5824) were obtained from ATCC. The SCLC cell line CORL311 was purchased from sigma (96020721). All the SCLC cell lines were maintained with ATCC-formulated RPMI-1640 medium (A1049101, Fisher Scientific) containing 10% FBS (MT35010CV, Sigma).

## Mouse experiments
All proposed research activities with vertebrate animals are conducted in the IACUC and AAALAC-approved Center for Comparative Medicine (CCM) facilities. This study was approved by Northwestern University Institutional Animal Care and Use Committee (Animal Protocol No. IS00013610). All animal studies were conducted in compliance with the ethical guidelines. Mice were housed 5 per cage and maintained under specific pathogen-free conditions. Twelve hours of light are provided each day. Food and water were provided freely. 5- to 6-week-old female athymic nude mice purchased from Envigo and were used for xenograft experiments. The human NCI-H526 SCLC cell line was injected into the right flank of nude mice ($1 \times 10^6$ cells per mice). Two weeks after inoculation, vehicle ($n = 7$) or 7.5 mg/kg of BRM014 ($n = 7$) was administered daily by intraperitoneal (IP) injections, and the tumor growth was measured every 2 days using a calibrated caliper. A two-tailed unpaired Student's t-test was used for statistical analysis. **$P < 0.01$; *$P < 0.05$.

## Immunoprecipitation (IP)
The IP experiment was performed as previously described[46]. Briefly, the cells were lysed in the lysis buffer (50 mM Tris pH 8.0, 150 mM NaCl, 0.5% Triton X100, 10% Glycerol, protease inhibitors, and benzonase). After centrifugation at 20,000 g at 4 °C for 15 min, the supernatants were collected and incubated with the antibody and immobilized Protein A/G (Santa Cruz) at 4 °C overnight with rotation. Then the Protein A/G beads were washed with ice-cold lysis buffer four times and boiled in 5× SDS sample loading buffer.

## CRISPR-mediated knockouts and Real-time PCR
Designed sgRNAs were cloned into either lentiCRISPR v2 (Addgene, 52961) lentiCRISPR v2-Blast (Addgene, 83480) vector. The cells were infected by lentivirus expressing targeting sgRNAs for 24 hours in the presence of 5 µg/ml polybrene. The cells were then selected by puromycin (2 µg/ml) for another two days. Oligo sequences used in this manuscript were as follows: sg*NONT* (GCTGAAAAAGGAAGGAGT TGA), sg*POU2AF2*–1 (GTGACGTCTACACCTCCAGCG), sg*POU2AF2*–2 (GAGAGGCAACTCGTGCTGGG), sg*POU2F3*–1 (GCCCACGCTTAGGGA-GATGTG), sg*POU2F3*–2 (GTCCTACCAAATACTTCACTG), sg*PTEN*–1 (AGCTGGCAGACCACAAACTG), sg*PTEN*–2 (ATTCTTCATACCAGGAC CAG). Primer sequences for Realtime PCR used in this manuscript were as follows: *FOXO1*_F: 5'-CTACGAGTGGATGGTCAAGAG-3', *FOXO1*_R: 5'-ATGAACTTGCTGTGTAGGGAC-3'; *IGFBP2*_F: 5'-ACATCCCCAACTGT-GACAAG-3', *IGFBP2*_R: 5'-ATCAGCTTCCCGGTGTTG-3'; *UBASH3B*_F: 5'-AAGCAAGACTAGTGGGTGAAG-3', *UBASH3B*_R: 5'-GGCTCTACACGGA TCTTCAAG-3'; *EYA1*_F: 5'-CCTTCCTCACAAACTATGGCTG-3', *EYA1*_R: 5'-ACCTTCAGTCTTGATGCCTG-3'; *TAS1R3*_F: 5'-GTTCTCTGTCTACG-CAGCTG-3', *TAS1R3*_R: 5'-TGGAAGGTCAGGTTGTACATG-3'; *IRAG2*_F: 5'-GAACACCGTCCCTCATTACC-3', *IRAG2*_R: 5'-TGGCTTCCTTGTCC TTTCTAC-3'; *AVIL*_F: 5'-CCATTATCAAGCCTACAGTCCC-3', *AVIL*_R: 5'-ATCATGGTTCAGTAAGTCCTGG-3'; *CHAT*_F: 5'-TGAGTACTGGCTGAA TGACATG-3', *CHAT*_R: 5'-AGTACACCAGAGATGAGGCT-3'; *SUCNR1*_F: 5'-GGATCAAGTCTTCCAACAGAATG-3', *SUCNR1*_R: 5'-TAGTACTTTTC-CAGGGCAGC-3'; *GNG13*_F: 5'-GAGAGCCTCAAGTACCAGC-3', *GNG13*_R: 5'-TGCCCTTTTCCACCCATG-3'; *NREP_F*: 5'-AAACAAGGACATGGAGGG AAG-3', *NREP_R*: 5'-GTGGAGGTAACTGATTCTTGGG-3'; *ASCL2_F*: 5'-CGTTCCGCCTACTCGTC-3', *ASCL2_R*: 5'-TGAGGCTCATAGGTCGA GG-3'.

## RNA-seq and analysis
RNA-seq was conducted as previously described[46]. Briefly, the cells were collected four days after CRISPR sgRNA treatment, and the total RNA was extracted with Qiagen RNeasy Plus Mini Kit (#74134). All the library construction steps were used according to the manufacturer's recommendations. Samples were pooled and sequenced on a NovaSeq with a read length configuration of 150 PE. Gene counts were computed by HTSeq v0.6.1[47] and used as an input for edgeR 3.0.8[48]. Genes with Benjamini-Hochburg adjusted p-values less than 0.01 were considered to be differentially expressed. RNA-seq heatmaps adjacent to ChIP-seq heatmaps display log2 (fold change) values of genes corresponding to nearest to ChIP-seq peaks and were displayed using Java TreeView v3.0. GO functional analysis was carried out using Metascape software v3.5[49] with default parameters.

## ChIP-seq Assay and analysis
ChIP-seq was performed as described previously[46]. Briefly, the cells were collected four days after CRISPR sgRNA treatment. Then the cell pellets were washed twice with ice-cold PBS and then fixed with 1% paraformaldehyde for 10 min at room temperature. Then the paraformaldehyde solution was quenched with 2.5 M glycine, and the cell pellets were washed twice with

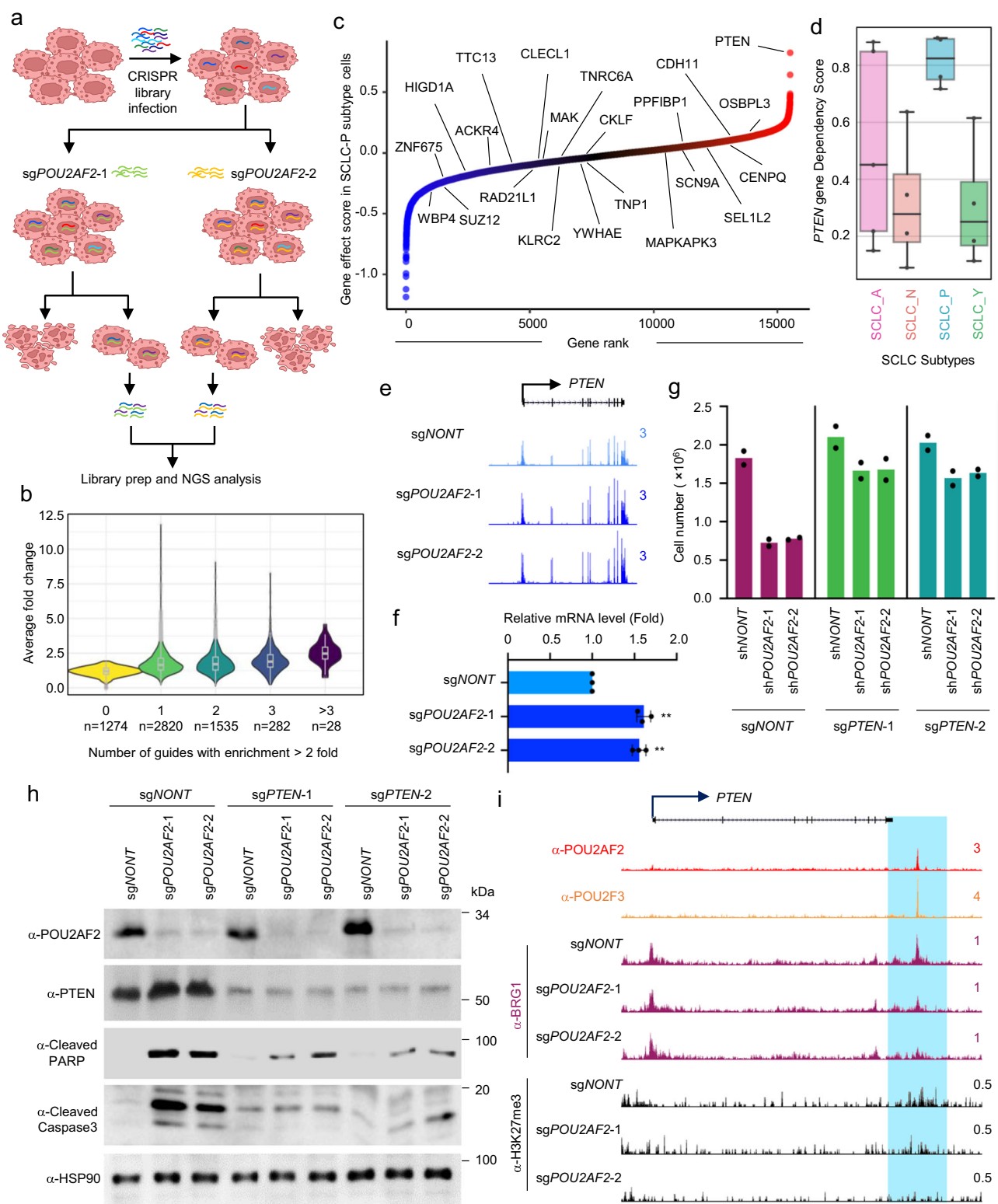

PBS afterwards. The cell pellets were resuspended with lysis buffer (50 mM HEPES, pH = 7.5, 140 mM NaCl, 1 mM EDTA, 10% Glycerol, 0.5% NP-40, 0.25% Triton X-100, 1× protease inhibitors) and incubated on nutator for 10 min in the cold room. Afterwards, cell pellets were centrifuged at 500 g for 5 min and discarded supernatant. Then, cell pellets were washed with wash buffer (10 mM Tris-HCl, pH = 8.0, 200 mM NaCl, 1 mM EDTA, 0.5 mM EGTA, 1 × protease inhibitors) and resuspended with the sonication buffer (10 mM Tris-HCl, pH = 8.0, 1 mM EDTA, 0.1% SDS, 1 ×

protease inhibitors). Sonication was performed with 1-mL Covaris tubes which were set to 5% to 10% duty factor, 175 peak intensity power, and 200 cycles per burst for 120-600 s. The 10× dilution buffer (10% Triton X-100, 1 M NaCl, 1% Na-Deoxycholate, 5% N-Lauroylsarcosine, 5 mM EGTA) was further added to the lysate, and samples were centrifuged at maximum speed for 15 min at 4 °C. The antibody was added (5 μg commercial antibody or 40 μl of homemade anti-sera) to each sample. After incubation at 4 °C overnight, 100 μl protein A/G agarose beads (Santa Cruz

**Fig. 5 | Loss of POU2AF2 results in SCLC cell viability decrease by activating PTEN expression. a** The flowchart shows genome-wide CRISPR-Cas9 screening with GeCKO lentiviral gRNA library in wild type and POU2AF2 depleted NCI-H526 cells. **b** The violin plot identifies the factors that depletion of which could rescue cell death induced by POU2AF2 depletion. X-axis represents the number of guides per gene identified in the screen (total 6 guides per gene). **c** The dependency score of the 28 genes that were identified from (**b**) in the four SCLC-P subtype cell lines: NCI-H526, NCI-H211, NCI-H1048, and CORL311. The gene effect score data were retrieved from the DepMap Portal. Center line: median; top and bottom hinges of box: the third and first quantiles; whiskers: quartiles ± 1.5 × interquartile range. **d** The dependency score of *PTEN* gene in the four SCLC subtypes (A, N, P, and Y). The following SCLC cell lines are used in the dependency analysis: A subtype, CORL47, DMS53, NCI-H1092, NCI-H209, SHP77 (*n* = 5); N subtype, CORL279, NCI-H1694, NCI-H466, NCI-H82 (*n* = 4); P subtype, CORL311, NCI-H1048, NCI-H211, NCI-H526 (*n* = 4); Y subtype, NCI-H1339, NCI-H2286, NCI-H841, SW1271 (*n* = 4). Center line:

median; top and bottom hinges of box: the third and first quantiles; whiskers: quartiles ± 1.5 × interquartile range. **e**) The track example shows the expression levels of *PTEN* gene in NCI-H526 cells transduced with either non-targeting sgRNA or two distinct POU2AF2 specific sgRNAs. **f** The mRNA levels of *PTEN* gene were determined in NCI-H526 cells transduced with either non-targeting sgRNA or two distinct POU2AF2 specific sgRNAs by real time PCR. *n* = 3 technical replicates. Data are presented as mean values ± standard deviation (SD). Source data are provided as a Source Data file. The NCI-H526 cells were transduced with lentivirus expressing either sg*NONT* or sg*POU2AF2* after PTEN depletion. The cell viability was determined by cell counting assay. *n* = 2 biologically independent experiments. Source data are provided as a Source Data file (**g**), and the protein levels of POU2AF2, PTEN, cleaved PARP, and cleaved Caspase 3 were determined by western blot. *n* = 2 biologically independent experiments. **h** Source data are provided as a Source Data file. **i** The track example shows the loss of BRG1 occupancy at the H3K27me3 marked enhancer region of *PTEN* gene locus.

Biotechnology, sc-2003) were added for 4 h. Then the beads were washed 4 times with RIPA buffer (50 mM HEPES, pH = 7.5, 500 mM LiCl, 1 mM EDTA, 1.0% NP-40, 0.7% Na-Deoxycholate), followed by once with ice-cold TE buffer (with 50 mM NaCl). The DNA for each IP sample was eluted with elution buffer (50 mM Tris-HCl, pH = 8.0, 10 mM EDTA, 1.0% SDS) and reverse cross-linked at 65 °C oven for 13 h, followed by protease K digestion at 55 °C for 2 h. The genomic DNA fragments were then further purified with Qiagen DNA purification kit (Qiagen, 28104).

For ChIP-seq data analysis, all the peaks were called with the MACS v2.1.0 software[50] using default parameters and corresponding input samples. Metaplots and heatmaps were generated by utilizing ngsplot v2.63 or deepTools v3.5.1 to display ChIP-seq signals. K-means clustering was also generated using ngsplot v2.63. Peak annotation was performed with ChIPseeker v1.38.0[51,52]. Motif analysis was performed with HOMER v4.11.

## ATAC-seq and analysis

ATAC-seq was performed as described previously[26]. Briefly, the frozen cells were thawed and washed once with PBS and then resuspended in cold ATAC lysis buffer. The cell number was then calculated by Cellometer Auto 2000 (Nexcelom Bioscience). For each sample, 50 K to 100 K nuclei were centrifuge (pre-chilled) at 500 *g* for 10 min. The supernatant was removed, and then the nuclei were resuspended in 50 µl of tagmentated DNA. The reactions were then incubated at 37 °C for 30 min in a thermomixer shaking at 1000 rpm, and then cleaned up by the MiniElute reaction clean up kit (Qiagen). Tagmented DNA was amplified with barcode primers. Library quality and quantity were assessed with Qubit 2.0 DNA HS Assay (ThermoFisher, Massachusetts, USA), Tapestation High Sensitivity D1000 Assay (Agilent Technologies, California, USA), and QuantStudio 5 System (Applied Biosystems, California, USA). Equimolar pooling of libraries was performed based on QC values and sequenced on an Illumina® HiSeq (Illumina, California, USA) with a read length configuration of 150 PE for 50 M PE reads (25 M in each direction) per sample. ATAC-seq reads are shifted + 4 bp and − 5 bp for positive and negative strands respectively using alignmentSieve function from deepTools package. ATAC-seq peaks are called with MACS v2.1.0.

## Hi-C and data processing

Hi-C samples were prepared with the Arima Hi-C kit according to the manufacturer's instruction. The adapters were trimmed from the Hi-C raw FASTQ files and the trimmed files were mapped against the hg19 human reference genome using the runHi-C pipeline (http://xiaotaowang.github.io/HiC_pipeline). Specifically, the Burrows-Wheeler Aligner was used for the FASTQ file alignment and aligned reads were filtered to remove low quality

reads and PCR duplicates. Aligned and filtered reads were then paired on the basis of read pairs and filtered to retain fragments that contain ligations of at least two different restriction fragments. These paired reads were then binned at 5-kb resolution. Hi-C contact matrix was generated at multiple resolutions (5 kb, 10 kb, 25 kb, 50 kb, 100 kb, 250 kb, 500 kb, 1 mb, 2.5 mb, and 5 mb) and normalized using ICE. AB compartments were identified at 100 kb resolution using cooltools (https://github.com/mirnylab/cooltools). The principal components values were derived using *eig-cis* from cooltools and the compartment A or compartment B were assigned based on the positive or negative PC1 value, respectively. A/B switch regions are defined as the regions with the sign of PC1 changed between sg*NONT* and *sgPOU2AF2*. To associate compartment switching with gene expression, we identified genes with at least 50% of gene body located within in the A/B switch regions and stable regions. The statistical significance of the difference in gene expression was tested by two-sided Wilcoxon test. Chromatin loops were called by Peakachu[53] at 10 kb resolution with probability score threshold greater than 0.98 from sg*NONT* and *sgPOU2AF2* Hi-C data. For differential loops, we calculated the probability score and compared the score between sg*NONT* and *sgPOU2AF*. sg*NONT* specific loops are defined as those with probability score > 0.98 in sg*NONT* and <0.5 in *sgPOU2AF*. *sgPOU2AF* specific loops are defined as those with probability score > 0.98 in *sgPOU2AF* and <0.5 in sg*NONT*. The APA plot was generated using the apa-analysis function of HiCPeaks (https://github.com/XiaoTaoWang/HiCPeaks).

## Genome-wide CRISPR screen

Genome wide CRISPR screen was performed as described previously[26]. Briefly, the SCLC cells were transduced with lentiviruses expressing the Human CRISPR Knockout Pooled Library[54] (Addgene 1000000048) for two weeks. The surviving cells were then infected by lentivirus expressing either non-targeting sgRNA or two distinct POU2AF2 specific sgRNAs. The cells were cultured for another two weeks before their genomic DNA was isolated and amplified with primers as described previously[26].

## Mass Spectrometry Sample Preparation and analysis

Mass spectrometry was performed as described previously[46]. Briefly, the protein pellet was denatured in 50 µL of 8 M Urea/ 0.4 M Ammonium Bicarbonate followed by reduction in 2 µL of 100 mM DTT. Then, the digests were acidified to 0.5% trifluoroacetic acid (TFA) and the peptides were desalted on C18 Sep-Paks (Waters). The pooled extracts were then dried in a vacuum concentrator and resuspended in 30 ul of 5% ACN/0.1% FA for LC-MS analysis.

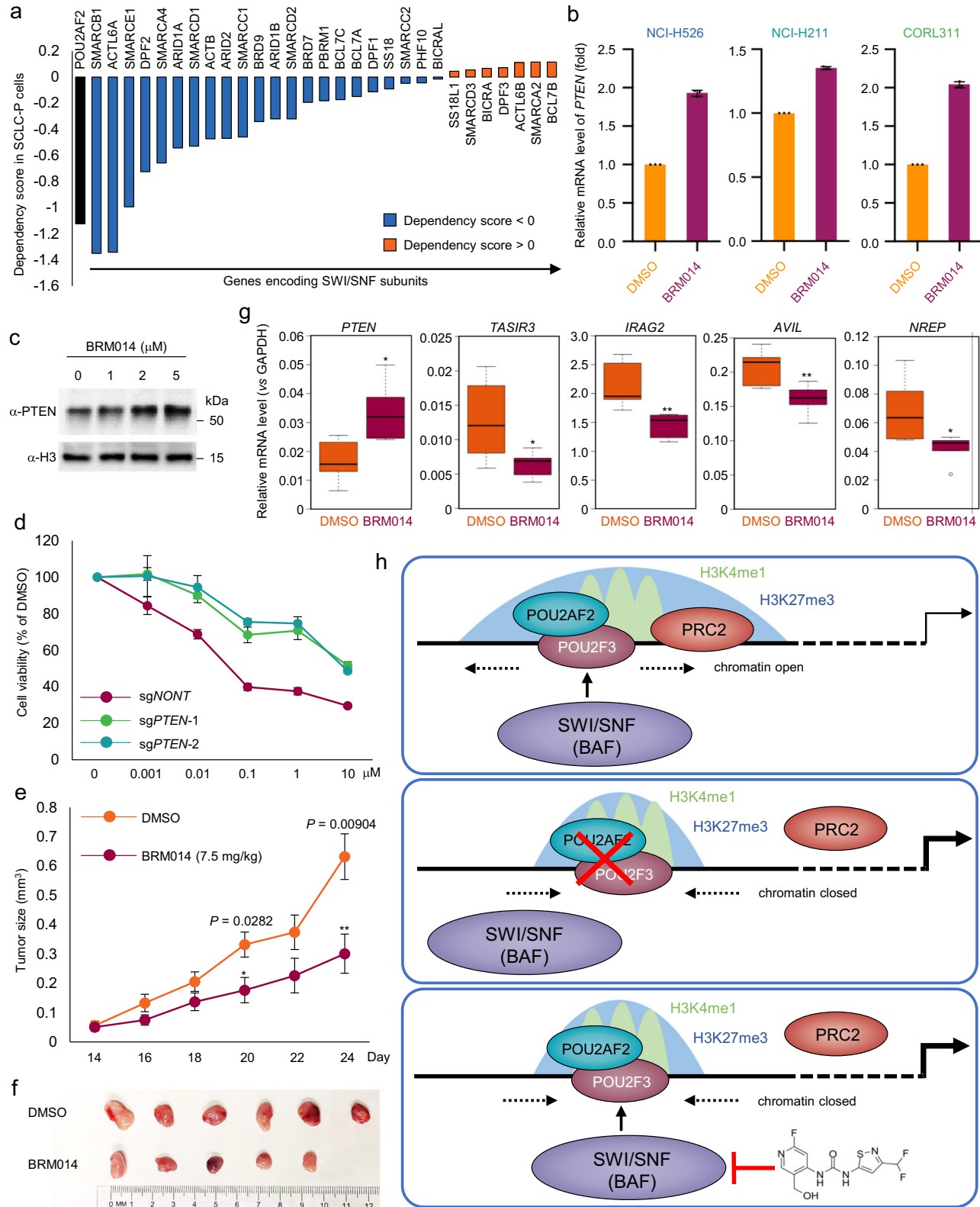

## Reporting summary

Further information on research design is available in the Nature Portfolio Reporting Summary linked to this article.

## Data availability

The raw and processed next-generation sequencing data generated in this study have been deposited to the Gene Expression Omnibus (GEO) database under the accession number GSE235704. The ChIP-seq data for POU2AF2, POU2F3, H3K4me1, H3K4me3, H3K27ac, and H3K27me3, as well as the gene expression data for POU2F3/POU2AF2 depletion and JQ1 treatment in NCI-H526 and NCI-H211 cells, were retrieved from our previous study (GSE197426)[11]. The mass spectrometry proteomics data have been deposited to the ProteomeXchange Consortium via the PRIDE partner repository

**Fig. 6 | Inhibition of the ATPase activity of SWI/SNF complex as a viable therapeutic strategy for SCLC treatment. a** The bar plot shows the gene dependency score (retrieved from DepMap database) of subunits of the SWI/SNF complex in NCI-H526 cells. **b** Three different SCLC-P cell line NCI-H526, NCI-H211, and CORL311 cells were treated with either DMSO or 1 μM BRM014 for 24 hours. The mRNA levels of *PTEN* were determined by real time PCR. *n* = 3 technical replicates. Data are presented as mean values ± standard deviation (SD). Source data are provided as a Source Data file. **c** The NCI-H526 cell line was treated with various concentrations of BRM014 for 24 hours. The protein levels of PTEN were determined by western blot. The total Histone H3 was used as internal control. *n* = 2 biologically independent experiments. Source data are provided as a Source Data file. **d** PTEN was depleted by two distinct sgRNAs in NCI-H526 cells. Then the cells were treated with different concentrations of BRM014 for 72 hours. The cell viability was determined by CellTiter-Glo Luminescent Cell Viability Assay. Data are presented as mean values ± standard deviation (SD). *n* = 3 biologically independent experiments. Source data are provided as a Source Data file. **e** $5.0 \times 10^6$ of NCI-H526 cells were inoculated into the right flank of nude mice. Two weeks after inoculation,

vehicle (*n* = 7) or 7.5 mg/kg of BRM014 (*n* = 7) was administered daily by intraperitoneal (IP) injections, and the tumor growth was measured every 2 days using a calibrated caliper. Data are represented as mean ± SEM. A two-tailed unpaired Student's t-test was used for statistical analysis. ** *P* < 0.01; * *P* < 0.05. Source data are provided as a Source Data file. **f** Images of representative tumors from each mouse were taken at the end of the experiment. Source data are provided as a Source Data file. **g** The total RNA was extracted from each tumor at the end of the experiment. The mRNA levels of *PTEN, TAS1R3, IRAG2, AVIL*, and *NREP* were determined by real time PCR. *n* = 3 technical replicates. A two-tailed unpaired Student's *t*-test was used for the statistical analysis. ** *p* < 0.01; * *p* < 0.05. Center line: median; top and bottom hinges of box: the third and first quantiles; whiskers: quartiles ± 1.5 × interquartile range. The *P*-value for gene expression difference between DMSO and BRM014 group was calculated as follows: *PTEN* (*P* = 0.011), *TAS1R3* (*P* = 0.0422), *IRAG2* (*P* = 0.0072), *AVIL* (*P* = 0.0097), and *NREP* (*P* = 0.0367). Source data are provided as a Source Data file. **h** The graphic model shows how POU2AF2 establishes the poised enhancer via SWI/SNF activity-dependent recruitment of the Polycomb complex in SCLC-P subtype cells.

with the dataset identifier PXD043238. All the remaining details and data associated with this study are available within the Methods section and the main text. Source data are provided with this paper.

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

## Acknowledgements

We are grateful to all the current and past Wang Lab members for their support. We want to thank Hiam Abdala-Valencia for CRISPR screen library sequencing. We thank Aidan Balagtas, Angela Beatrice Lichauco, and Rima Tulaiha for the experimental material prep. The lentiCRISPR v2 vector was a gift from Feng Zhang. The lentiCRISPR v2-Blast vector was a gift from Mohan Babu. L.W. is supported by NIH grant R35GM146979, the Research Scholar Grant (RSG-22-039-01-DMC) from the American Cancer Society, and the Idea Development Award (HT94252310360) from USAMRAA. F.Y. is supported by NIH grants R35GM124820, R01HG009906, and R01HG011207.

## Author contributions

L.W. and Z.Z. designed the study. A.S., N.T., and L.W. performed all the biochemical and sequencing experiments. H.L., P.W., and F.Y. performed and analyzed the Hi-C experiment. B.D.S. performed the CRISPR library sequencing. Z.Z. performed the other bioinformatic analysis. A.S. and L.W. wrote the manuscript. O.B., T. Z., Z.Z., and L.W. revised the manuscript. All authors read and approved the final manuscript.

## Competing interests

The authors declare no competing interests.
