## [Peer Review File · Nature Communications]

A SWI/SNF-dependent transcriptional regulation mediated by POU2AF2/C11orf53 at enhancerEditorial Note: Parts of this Peer Review File have been redacted as indicated to remove third-party material where no permission to publish could be obtained.

REVIEWER COMMENTS

Reviewer #1 (Remarks to the Author):

This manuscript from Szczepanski reports that master transcription factors POU2F3 and POU2AF2 have an uncharacterized role as transcriptional repressors in tuft cells and small cell lung cancer (SCLC). These transcription factors (TFs) were recently shown by several groups to play a critical role in maintaining tuft cell identity, with loss of either POU2F3 or POU2AF2 in SCLC cell lines causing a loss of expression of tuft cell identity genes and reduction in cell proliferation. Given the binding of POU2F3 and POU2AF2 to regions enriched in H3K27ac and Mediator, and the similarity of POU2AF2 to the activator OCA-B, these proteins were reported to be activators of tuft-cell transcription programs, and to support tuft cell state.

Here, the authors report that POU2F3 and POU2AF2 also occupy a number of sites (Cluster 3) that lack H3K27ac and are marked instead with H3K27me3, which they refer to as poised enhancers. The authors state that the genes nearest these poised enhancers are upregulated upon depletion of POU2F3 or POU2AF2, and suggest that POU2F3 and POU2AF2 might directly repress expression of these genes. The authors present evidence that depletion of POU2AF2 might reduce Polycomb occupancy of the selected sites, thereby allowing the nearby genes to be expressed. However, whether the modest changes observed in Polycomb distribution or gene expression are direct effects of depleting POU2F3 or POU2AF2, or secondary consequences of disrupting tuft cell identity remain unclear.

The authors perform mass spec with GFP-tagged POU2AF2, and fail to detect members of the Polycomb complexes, but do find several subunits of the SWI/SNF complex. Thus, like other POU family TFs, POU2F3/POU2AF2 appear to cooperate with SWI/SNF to generate open chromatin at POU2F3/POU2AF2-bound lineage-specific enhancers (see for example, data from the Schubleler and Klose labs on OCT4). As anticipated, inhibiting SWI/SNF caused these lineage specific enhancers to lose chromatin accessibility and active enhancer marks, as well as reducing the expression of POU2F3/POU2AF2 target genes. Notably, this treatment also caused a modest change in the distribution of Polycomb away from the Cluster 3 sites, and increased expression of Cluster 3 genes.

The authors also perform a CRISPR screen for genes that can rescue cell death upon POU2AF2 KO, and find PTEN. They propose that POU2F3/POU2AF2 actively represses PTEN in SCLC cells, suggesting that the POU2AF2/PTEN axis could be a potential therapeutic target in SCLC-P subtype disease.

Overall, I find the main message of this work - that POU2F3/POU2AF2 can act as a transcriptional repressor of a set of cluster 3 genes - to be poorly supported. Major concerns are listed below:

1) The authors should show more clearly the expression level of genes near the 'poised enhancers' bound by POU2F3 and POU2AF2 (Cluster 3, Figure 1A) upon depletion of POU2F3 or POU2AF2. The heatmap shown in Figure 1D should be accompanied by violin plots showing the expression of these genes in each condition (sgNT, sgPOU2Fs, sgPOU2AF2), with some measure of significance. The scale on Figure 1D maxes out at Log2FC of 0.5, and most genes in Cluster 3 look white or very light yellow, making it seem that a majority of these genes are not substantially upregulated when POU2F3 or POU2AF2 are depleted.

2) The relevance of the JQ1 experiments to the story is not evident. The authors claim that these experiments support that activation of Cluster 3 genes might not be a secondary effect of the repression of Cluster 2 genes, but this argument is not compelling as presented. It still seems most likely that the gene activation observed is an indirect, secondary effect of cell state changes caused by depleting master TFs.

3) The authors state that 'the ChIP-seq assay revealed a significant reduction in H3K27me3 levels at

Cluster 3 regions', but I couldn't find any statistical test that supported significance. In fact, when I looked through the methods and data reporting summary, I found that each ChIP-seq experiment had been performed only once. Further, it appears that each of these single replicates had only been sequenced to 5 million reads. Very low sequencing depth with no replicate samples for any of the ChIP-seq or ATAC-seq experiments seems to me inadequate to support the claims made in this manuscript.

4) There were no experimental details given for the ChIP-seq. The methods section reads: 'ChIP-seq was performed as described previously', with no citation. This is not sufficient information. ATAC-seq, RNA-seq, Immunoprecipitations, etc. are all similarly lacking any information to tell a reader how the experiments were performed.

5) The fact that inhibition of SWI/SNF ATPases causes increased expression of Cluster 3 genes suggests that the repression of POU2F3/POU2AF2-bound lineage-specific enhancers by any means has secondary, indirect effects on the expression of Cluster 3 genes. To my mind, these data argue against the author's conclusions that POU2F3/POU2AF2 plays a direct repressive role. If the authors see this differently, it would be helpful for them to explain their rationale?

6) The authors imply that loss of POU2AF2 activates PTEN, but don't show this directly despite having RNA-seq data under these conditions (for example, by showing RNA-seq tracks at this gene). Because I found this concerning, I looked in the publicly available data from Chris Vakoc's lab (PMCID: PMC9419707), where they performed RNA-seq in 4 different SCLC cell lines, comparing mRNA changes after POU2AF2 (3 different sgRNAs) or POU2F3 (2 different sgRNAs) knockout compared with control sgRNAs (2 different sgRNAs). RNA was collected 5 days after sgRNA infection for NCI-H211, or 6 days after infection with sgRNA for NCI-H526, COR-L311 and NCI-H1048 cells. In their data, PTEN was not significantly upregulated following sgRNA treatment for POU2F3 or POU2AF2 KO in any of the four cell lines. To understand the basis of this discrepancy, I looked for the duration of sgRNA infection, number of replicates, cell line growth information, etc. in the current manuscript, but couldn't find any of this information. I am thus unable to assess the data in this manuscript, or why it is so different from the comprehensive and rigorous work published by the Vakoc group.

7) The experiments in this manuscript to get at POU2F3 or POU2AF2 function are all performed after days of TF knockdown, opening the door to indirect and secondary effects. Given that several groups have shown loss of cell viability and cell state within several days of depletion of POU2F3 or POU2AF2 in SCLC cell lines, one would immediately assume that many genes would be altered (up and down) as the cells totally alter their cell state. Thus, a study of the direct effects of these master TFs should employ rapid depletion (e.g., degron) strategies, or do a better job acknowledging the caveats that the effects observed could be fully indirect.

Reviewer #3 (Remarks to the Author):

In this manuscript, the authors demonstrate that POU2AF2, a newly identified co-factor of lineage-specific transcription factor POU2F3 in small cell lung cancer (SCLC), marks not only active enhancers, but also a set of PRC2 target genes. Mechanistically, the authors revealed that POU2AF2 recruits SWI/SNF chromatin remodeling complex to PRC2 sites to facilitate the maintenance of PRC2 occupancy and subsequent silencing of target genes. The authors further conducted a CRISPR screen and identified a critical role of PTEN suppression through maintaining POU2AF2-mediated PRC2 occupancy in SCLC cancer cell growth. Lastly, the authors provided the evidence that pharmacologic targeting of the SWI/SNF ATPase activities suppressed the SCLC-POU2F3 subtype tumor growth mainly through suppressing PTEN expression. This study is well conducted and indicates that while SWI/SNF has a wide-spread remodeling chromatin activity to activate lineage-specific enhancers, it

can also recruit PRC2 to suppress tumor suppressor genes in certain cell context.

Major comments:

1. The authors suggested that increased chromatin accessibility enables the recruitment of PRC2. However, it is unclear how cells choose PRC2 rather than HATs to inactivate enhancers of cluster 3 genes.
2. Missing some details in figure legends/text. For example, were data presented in Fig.2A/B and Fig.S2B from the same experiment? While Fig. 2A/B suggested a significant overlapping, Fig. S2B showed only a partial overlapping with 195 out of 926 POU2AF2 KO-upregulated genes being upregulated by GSK126 treatment. This may suggest that while POU2AF2 is essential for PRC2 occupancy, other mechanisms may be involved in the regulation of POU2AF2 KO-upregulated genes. The authors should comment on this.
3. The authors performed BRG1 ChIP-seq in POU2AF2-depleted cells and identified that both the retained and lost peaks of BRG1 were significantly enriched with the POU2F3 motif. It was further demonstrated that POU2AF2 depletion impaired BRG1 retention on cluster 2/3 genomic loci, but not those of cluster 1 genes. Given that they all have POU2F3 motifs, it is unclear why POU2AF2 loss only impaired cluster 2/3 genes. Does it suggest POU2AF2 function independently of POU2F3 binding?
4. While it is logical to target SWI/SNF to activate PTEN in SCLC-P, is the effect specific to this subtype? In addition, SWI/SNF ATPase inhibitors have not shown promise in clinical testing, which limits the translation potential of this study. Based on the model proposed by the authors, targeting PRC2 would activate PTEN. As EZH2 inhibitors have been approved by FDA, it will have a rapid translation for clinical testing.

Minor points:

1. Fig.3. the authors states that "depletion of POU2AF2 by two distinct sgRNAs significantly reduced the chromatin occupancy of the vast majority of BRG1 peaks at the Cluster 2 and Cluster 3 peaks, but less so at Cluster 1 (Fig. 3H, I; Fig. S3F, G)". A statistical analysis is missing to support this statement. In addition, the track example should be clearly described in the text. There is also a labeling error for last two panels (Panels J and K should be I and J).

Reviewer #1 (Remarks to the Author):

This manuscript from Szczepanski reports that master transcription factors POU2F3 and POU2AF2 have an uncharacterized role as transcriptional repressors in tuft cells and small cell lung cancer (SCLC). These transcription factors (TFs) were recently shown by several groups to play a critical role in maintaining tuft cell identity, with loss of either POU2F3 or POU2AF2 in SCLC cell lines causing a loss of expression of tuft cell identity genes and reduction in cell proliferation. Given the binding of POU2F3 and POU2AF2 to regions enriched in H3K27ac and Mediator, and the similarity of POU2AF2 to the activator OCA-B, these proteins were reported to be activators of tuft-cell transcription programs, and to support tuft cell state.

Here, the authors report that POU2F3 and POU2AF2 also occupy a number of sites (Cluster 3) that lack H3K27ac and are marked instead with H3K27me3, which they refer to as poised enhancers. The authors state that the genes nearest these poised enhancers are upregulated upon depletion of POU2F3 or POU2AF2, and suggest that POU2F3 and POU2AF2 might directly repress expression of these genes. The authors present evidence that depletion of POU2AF2 might reduce Polycomb occupancy of the selected sites, thereby allowing the nearby genes to be expressed. However, whether the modest changes observed in Polycomb distribution or gene expression are direct effects of depleting POU2F3 or POU2AF2, or secondary consequences of disrupting tuft cell identity remain unclear.

The authors perform mass spec with GFP-tagged POU2AF2, and fail to detect members of the Polycomb complexes, but do find several subunits of the SWI/SNF complex. Thus, like other POU family TFs, POU2F3/POU2AF2 appear to cooperate with SWI/SNF to generate open chromatin at POU2F3/POU2AF2-bound lineage-specific enhancers (see for example, data from the Schubleler and Klose labs on OCT4). As anticipated, inhibiting SWI/SNF caused these lineage specific enhancers to lose chromatin accessibility and active enhancer marks, as well as reducing the expression of POU2F3/POU2AF2 target genes. Notably, this treatment also caused a modest change in the distribution of Polycomb away from the Cluster 3 sites, and increased expression of Cluster 3 genes.

The authors also perform a CRISPR screen for genes that can rescue cell death upon POU2AF2 KO, and find PTEN. They propose that POU2F3/POU2AF2 actively represses PTEN in SCLC cells, suggesting that the POU2AF2/PTEN axis could be a potential therapeutic target in SCLC-P subtype disease. Overall, I find the main message of this work - that POU2F3/POU2AF2 can act as a transcriptional repressor of a set of cluster 3 genes - to be poorly supported. Major concerns are listed below:

- 1) The authors should show more clearly the expression level of genes near the 'poised enhancers' bound by POU2F3 and POU2AF2 (Cluster 3, Figure 1A) upon depletion of POU2F3 or POU2AF2. The heatmap shown in Figure 1D should be accompanied by violin plots showing the expression of these genes in each condition (sgNT, sgPOU2Fs, sgPOU2AF2), with some measure of significance. The scale on Figure 1D maxes out at Log₂FC of 0.5, and most genes in Cluster 3 look white or very light yellow, making it seem that a majority of these genes are not substantially upregulated when POU2F3 or POU2AF2 are depleted.

We thank the reviewer for pointing this out. Following the reviewer's suggestions, we conducted

an additional statistical analysis to measure the significance of the up-regulation of Cluster 3-related genes. As shown in Figure S1E of the revised manuscript, we detected a significant increase in Cluster 3 gene expression (with the y-axis representing the log₁₀ scale of RPM) in cells transduced with two distinct POU2AF2 sgRNAs. For the scale bar in Figure 1D that the reviewer referred to, we used Log₂FC of 0.5 in order to cover most of the genes. However, Log₂FC 0.5 is not the maximum change in gene expression. As shown in Figure 1B, there are numbers of genes up-regulated more than 2-fold (log₂FC >1) after POU2AF2 depletion.

Fig. S1e

2) The relevance of the JQ1 experiments to the story is not evident. The authors claim that these experiments support that activation of Cluster 3 genes might not be a secondary effect of the repression of Cluster 2 genes, but this argument is not compelling as presented. It still seems most likely that the gene activation observed is an indirect, secondary effect of cell state changes caused by depleting master TFs.

We thank the reviewer for pointing this out. First, we determined the expression levels of POU2F3/POU2AF2 targeted tuft cell-specific genes in cells treated with either DMSO or JQ1. As shown in Panel A below, all these cell type-specific genes were significantly reduced upon JQ1 treatment based on the RNA-seq analysis. Then, we conducted box-plot analysis to determine the global gene expression change upon JQ1 treatment in each cluster. As shown in Panel B below, there is also a significant decrease in Cluster 3 genes. This result suggests that the loss of tuft cell-specific gene expression signatures will not directly lead to a global increase of Cluster 3 genes. This is a major difference between the effect of losing POU2AF2/POU2F3 and treatment with JQ1 on the impact on Cluster 3 gene expression. However, we agree with the reviewer that it is also possible that the change in some of the Cluster 3 genes may be due to the secondary effect of Cluster 1/2 genes or cell type/viability change; therefore, we provided an additional discussion in the revised manuscript to address this concern. We have attached the figure here for the reviewer's convenience.

3) The authors state that ‘the ChIP-seq assay revealed a significant reduction in H3K27me3 levels at Cluster 3 regions’, but I couldn’t find any statistical test that supported significance. In fact, when I looked through the methods and data reporting summary, I found that each ChIP-seq experiment had been performed only once. Further, it appears that each of these single replicates had only been sequenced to 5 million reads. Very low sequencing depth with no replicate samples for any of the ChIP-seq or ATAC-seq experiments seems to me inadequate to support the claims made in this manuscript.

We thank the reviewer for pointing out this typo in our original reporting summary. All our ChIP-seq results contain at least 30 million reads, which has been revised in our reporting summary. Following the reviewer’s suggestion, we repeated the ChIP-seq experiments and included biological replicates and statistical analysis as shown in Figures S1E, S2D, S2F, 3C, 3H, S3F, S3G, S3H, S4G, and 5F in the revised manuscript. In particular, for H3K27me3 and EZH2 ChIP-seq in Figure 2E, we provide the statistical analysis for both replicates. As shown in Figure S2D, there is a significant reduction of EZH2 levels at Cluster 3 loci after POU2AF2 depletion. We have attached the figure here for the reviewer’s convenience.

Fig. S2d

4) There were no experimental details given for the ChIP-seq. The methods section reads: ‘ChIP-seq was performed as described previously’, with no citation. This is not sufficient information. ATAC-seq, RNA-seq, Immunoprecipitations, etc. are all similarly lacking any information to tell a reader how the experiments were performed.

We thank the reviewer for making this excellent point. We have now provided the correct citations and detailed experimental descriptions in the revised manuscript. We have highlighted the information in the revised manuscript for the convenience of the reviewer.

5) The fact that inhibition of SWI/SNF ATPases causes increased expression of Cluster 3 genes suggests that the repression of POU2F3/POU2AF2-bound lineage-specific enhancers by any means has secondary, indirect effects on the expression of Cluster 3 genes. To my mind, these data argue against the author's conclusions that POU2F3/POU2AF2 plays a direct repressive role. If the authors see this differently, it would be helpful for them to explain their rationale? We thank the reviewer for making this excellent point. As discussed in our manuscript, it has been well-documented that the SWI/SNF complex has opposing functions in transcriptional regulation (either transcriptional activation or repression). For instance, loss of the core subunit SNF5 in SWI/SNF would lead to elevated expression of EZH2 and strong repression of Polycomb targets in MEF cells (*Cancer Cell*. 2010 Oct 19; 18(4): 316-28). In B cells, the SWI/SNF complex is responsible for transcriptional repression via interaction with transcriptional co-repressors, and loss of BRG1 reduces H3K27me3 levels and depresses gene expression (*Proc Natl Acad Sci U S A*. 2015 Feb 17; 112(7): E718-27). In a more recent study from Dr. Crabtree's group (*Nat Struct Mol Biol*. 2021 Jun; 28(6): 501-511), the authors demonstrated that once the SWI/SNF is rapidly degraded, Polycomb complexes may accumulate and redistribute away from heavily occupied sites, leading to PRC redistribution, physical decompaction of the genome, accumulation of active marks, and derepression. Similarly, in our current study, we found that the SWI/SNF complex, which is recruited by POU2AF2, also functions as a transcriptional repressor at Polycomb targets in Cluster 3. In fact, it has been reported before that BRG1 (ATPase of SWI/SNF) is required to maintain Polycomb-mediated repression of non-mesodermal developmental regulators during the mesoderm induction (*Development*. 2015 Apr 15;142(8):1418-30). The switch made between the transcriptional active and repressive functions of POU2AF2/POU2F3 or the SWI/SNF complex may be due to context-specific epigenetic states of the chromatin, which is a major question in this field and warrants further investigation. We also agree with the reviewer that the transcriptional repressive function of POU2AF2/POU2F3 may be interconnected with its active function at the chromatin. Therefore, to make our conclusion more accurate, we revised the overstatement that POU2AF2/POU2F3 functions as a direct transcriptional repressor in the revised manuscript.

6) The authors imply that loss of POU2AF2 activates PTEN, but don't show this directly despite having RNA-seq data under these conditions (for example, by showing RNA-seq tracks at this gene). Because I found this concerning, I looked in the publicly available data from Chris Vakoc's lab (PMCID: PMC9419707), where they performed RNA-seq in 4 different SCLC cell lines, comparing mRNA changes after POU2AF2 (3 different sgRNAs) or POU2F3 (2 different sgRNAs) knockout compared with control sgRNAs (2 different sgRNAs). RNA was collected 5 days after sgRNA infection for NCI-H211, or 6 days after infection with sgRNA for NCI-H526, COR-L311 and NCI-H1048 cells. In their data, PTEN was not significantly upregulated following sgRNA treatment for POU2F3 or POU2AF2 KO in any of the four cell lines. To understand the basis of this discrepancy, I looked for the duration of sgRNA infection, number of replicates, cell line growth information, etc. in the current manuscript, but couldn't find any of this information. I am thus unable to assess the data in this manuscript, or why it is so different from the comprehensive and rigorous work published by the Vakoc group.

We thank the reviewer for pointing out this. We have shown the tracks of the *PTEN* gene in cells

transduced with either non-targeting sgRNA or two distinct POU2AF2 sgRNAs in Figure 5E of our manuscript. In the revised manuscript, we further provided the qPCR validation as shown in Figure 5F.

We thank the reviewer for noticing the difference between our findings and the published work by the Vakoc group. As the reviewer pointed out, Dr. Vakoc's group conducted the RNA-Seq experiment 6 days after the sgRNA infection. However, in our current work and our previously published work (*Sci Adv.* 2022 Oct 7; 8(40): eabq2403), we collected the total RNA for RNA-seq 4 days after the sgRNA infection. The detailed experiment conditions are included in our revised method section. We chose this time point because we noticed that depletion of POU2AF2 led to a strong inhibition of cell growth and viability starting from Day 4, as we reported before (please also refer to the left panel below). In support of our work, in the back-to-back paper published by Dr. Yang Shi's group (*Cell Discov.* 2022 Oct 18;8(1):112), they also reported a significant reduction (>90%) of cell viability after 5-day sgRNA treatment or 4-day shRNA treatment (please refer to the right panel below). Therefore, we chose the 4-day time point for all of the NGS experiments in order to detect the earlier transcriptional alteration upon POU2AF2 depletion. It is possible that most of the PTEN-sensitive cells die after 5-6 days of depletion of POU2AF2, and the surviving cells may be resistant to POU2AF2 depletion-induced PTEN upregulation.

[redacted]

In addition, we analyzed the ChIP-seq data (GSE186614) published by Dr. Vakoc's group in their *Nature* paper to determine whether POU2F3 and POU2AF2 could be detected at the *PTEN* locus in their system. As shown in the figure below, their antibodies could also detect the POU2F3/POU2AF2 occupancy as at the indicated *PTEN* locus as our antibodies.

Since both of the antibodies detected POU2F3/POU2AF2 occupying the *PTEN* locus, we further analyzed the RNA-seq from both the *Genes & Dev* (PMC6075037) and *Nature* paper (PMC9419707) published by Dr. Vakoc's group to determine whether the expression levels of *PTEN* have changed after POU2F3 or POU2AF2 depletion. Based on the analysis with their published data, there is a significant increase in *PTEN* expression in POU2F3-depleted cells in both papers. However, the expression of *PTEN* in POU2AF2-depleted cells tends to increase but not significantly (please kindly refer to the table below). Therefore, another possibility is that the efficacy of their POU2AF2 sgRNAs is different from POU2F3 sgRNAs.

	Log2FC (vs. sgNEG)	adj.p	GEO	Publication
sgPOU2F3	0.648401247	5.47E-10	GSE115122	Genes & Development
sgPOU2F3	0.601350701	7.19E-07	GSE186614	Nature
sgPOU2AF2	0.371110477	0.05535	GSE186614	Nature

7) The experiments in this manuscript to get at POU2F3 or POU2AF2 function are all performed after days of TF knockdown, opening the door to indirect and secondary effects. Given that several groups have shown loss of cell viability and cell state within several days of depletion of POU2F3 or POU2AF2 in SCLC cell lines, one would immediately assume that many genes would be altered (up and down) as the cells totally alter their cell state. Thus, a study of the direct effects of these master TFs should employ rapid depletion (e.g., degron) strategies, or do a better job acknowledging the caveats that the effects observed could be fully indirect.

We thank the reviewer for making this excellent point. It has been well documented that transcription factors can either repress or activate the transcription of target genes by recruiting other cofactors (PMID: 11295539), such as the Class II POU domain transcription factors (POU2F1, POU2F2, and POU2F3) (PMID: 23580612, 26271992, 21051540). For instance, in colon cancer cells, POU2F1 promotes a repressed state of gene expression through recruiting the NuRD chromatin-remodeling complex (PMID: 21051540). In breast cancer cells, both POU2F1 and POU2F2 could bind to the promoter of *iNOS* gene, forming a higher-order complex which fails to recruit RNA polymerase (PMID: 26271992). In our system, we discovered that POU2F3/POU2AF2 is critical for the chromatin recruitment of the SWI/SNF complex, which subsequently maintains the function of PRC2 at inactive enhancer elements. We also agree with the reviewer that there is no evidence showing that POU2AF2 or POU2F3 could directly repress gene expression since neither one contains any known transcriptional repressive domain. Due to

the technical difficulties of making AID-tag knock-in in the SCLC cell aggregates, we rephrased our statement, acknowledging the possibility of an indirect effect following the reviewer's suggestion.

Reviewer #3 (Remarks to the Author):

In this manuscript, the authors demonstrate that POU2AF2, a newly identified co-factor of lineage-specific transcription factor POU2F3 in small cell lung cancer (SCLC), marks not only active enhancers, but also a set of PRC2 target genes. Mechanistically, the authors revealed that POU2AF2 recruits SWI/SNF chromatin remodeling complex to PRC2 sites to facilitate the maintenance of PRC2 occupancy and subsequent silencing of target genes. The authors further conducted a CRISPR screen and identified a critical role of PTEN suppression through maintaining POU2AF2-mediated PRC2 occupancy in SCLC cancer cell growth. Lastly, the authors provided the evidence that pharmacologic targeting of the SWI/SNF ATPase activities suppressed the SCLC-POU2F3 subtype tumor growth mainly through suppressing PTEN expression. This study is well conducted and indicates that while SWI/SNF has a wide-spread remodeling chromatin activity to activate lineage-specific enhancers, it can also recruit PRC2 to suppress tumor suppressor genes in certain cell context.

Major comments:

1. The authors suggested that increased chromatin accessibility enables the recruitment of PRC2. However, it is unclear how cells choose PRC2 rather than HATs to inactivate enhancers of cluster 3 genes.

We thank the reviewer for making this excellent point. As we show in Figure 1A, the H3K27ac level is very low at Cluster 3 loci. However, depletion of POU2AF2 activates gene expression in an H3K27me3-dependent manner without increasing H3K27ac levels. Therefore, we conclude that the HATs or HDACs may not be involved in POU2AF2-mediated transcriptional repression at the inactive enhancers (Cluster 3) and decided to focus on PRC2 function in our studies.

To extend further upon the reviewer's point, we considered the role of DNA methylation to possibly regulate Cluster 3 gene expression. DNA methylation is one of the major mechanisms that mediates HDACs' recruitment to chromatin. Therefore, we conducted whole genome bisulfate sequencing (WGBS) in H526 cells to determine the DNA methylation levels within the Cluster 3 regions (please refer to the figure below). We found very low levels of DNA methylation in Cluster 3 regions, suggesting that HDACs may not be recruited by POU2AF2 to repress gene expression. In fact, POU2F3/POU2AF2 binds to the AT-rich motif (5'-ATTTGCAT-3'), which may not be regulated by DNA methylation. We discussed more on this point in the revised manuscript, and we aim to further address the question as to why POU2AF2 cofunctions with PRC2 but not HDACs or HATs in our future studies.

2. Missing some details in figure legends/text. For example, were data presented in Fig.2A/B and Fig.S2B from the same experiment? While Fig. 2A/B suggested a significant overlapping, Fig. S2B showed only a partial overlapping with 195 out of 926 POU2AF2 KO-upregulated genes being upregulated by GSK126 treatment. This may suggest that while POU2AF2 is essential for PRC2 occupancy, other mechanisms may be involved in the regulation of POU2AF2 KO-upregulated genes. The authors should comment on this.

We thank the reviewer for making this excellent point. Figures 2A, 2B, and S2B are generated from the same RNA-seq experiments, and we included this information in the revised figure legend. Figure 2A overlapped all the co-up-regulated genes by GSK126 treatment and POU2AF2 depletion, while Figure S2B only shows the overlapped genes with $|\log_2FC| > 1$. We agree with the reviewer that other mechanisms may be involved in the regulation of POU2AF2 KO-upregulated genes, and therefore, we provided additional discussion in the revised manuscript. We also highlighted the discussion for the reviewer's convenience.

3. The authors performed BRG1 ChIP-seq in POU2AF2-depleted cells and identified that both the retained and lost peaks of BRG1 were significantly enriched with the POU2F3 motif. It was further demonstrated that POU2AF2 depletion impaired BRG1 retention on cluster 2/3 genomic loci, but not those of cluster 1 genes. Given that they all have POU2F3 motifs, it is unclear why POU2AF2 loss only impaired cluster 2/3 genes. Does it suggest POU2AF2 function independently of POU2F3 binding?

We thank the reviewer for pointing out this. In Figure 3C, we centered all the peaks on total BRG1 peaks ($n = 58,182$) and found that loss of POU2AF2 leads to a redistribution of total BRG1 peaks. Although we detected an enrichment of the POU2F3 motif in both lost and retained peaks, we noticed that the p-value is quite different, suggesting that lost BRG1 peaks are more enriched at *POU2F3* loci (please kindly refer to the figure below). Because we are not able to completely deplete POU2AF2 or POU2F3 from SCLC-P cells, the residual POU2AF2/POU2F3 proteins may still be able to mediate BRG1 recruitment.

To address the second question by the reviewer, depletion of POU2AF2 actually leads to a reduction of BRG1 in all three clusters, based on the analysis with two biological replicates (please kindly refer to the figure below). Therefore, we rephrased the statement in the revised manuscript. Finally, we agree with the reviewer that POU2AF2 may have an POU2F3-independent function because POU2AF2 could also be detected in the cytoplasm while POU2F3 is exclusively localized in the nuclei. We have been actively working on elucidating the POU2F3-independent function of POU2AF2 and will show the results in our future studies.

Fig. S3G

Fig. S3H

4. While it is logical to target SWI/SNF to activate PTEN in SCLC-P, is the effect specific to this subtype? In addition, SWI/SNF ATPase inhibitors have not shown promise in clinical testing, which limits the translation potential of this study. Based on the model proposed by the authors, targeting PRC2 would activate PTEN. As EZH2 inhibitors have been approved by FDA, it will have a rapid translation for clinical testing.

We thank the reviewer for making this excellent point. We don't think activating PTEN by targeting SWI/SNF is specific to SCLC-P subtype only, because it has been published before that Brg1 depletion could lead to the activation of PTEN in colorectal carcinoma cells (PMC3039810). However, no previous studies have shown the effect of SWI/SNF inhibitor on SCLC cells. Based on the dependency analysis (Fig. 6A), we found the SCLC-P subtype cells are very sensitive to SWI/SNF depletion, and therefore, we tested the antitumor effect of SWI/SNF inhibitor in our animal models. We agree with the reviewer that there might be a translational limit for SWI/SNF inhibitors since SWI/SNF mainly functions as tumor suppressors in some cancer types, and EZH2 inhibitors might be a better option and it has been published that EZH2 occupied *PTEN* loci and inhibit *PTEN* expression in gastric cancer (PMC5769437). Following the reviewer's suggestion, we provided additional discussion in the revised manuscript and highlighted in the manuscript for the reviewer's convenience.

Minor points:

1. Fig.3. the authors states that “depletion of POU2AF2 by two distinct sgRNAs significantly reduced the chromatin occupancy of the vast majority of BRG1 peaks at the Cluster 2 and Cluster 3 peaks, but less so at Cluster 1 (Fig. 3H, I; Fig. S3F, G)”. A statistical analysis is missing to support this statement. In addition, the track example should be clearly described in the text. There is also a labeling error for last two panels (Panels J and K should be I and J). We thank the reviewer for pointing out this. Following the reviewer's suggestion, we provided the p-value in the revised manuscript (Fig. S3H; please kindly refer to the figure below). Based on Reviewer #1's suggestion, we also provided a biological replicate for BRG1 ChIP-seq in cells transduced with either non-targeting sgRNA or POU2AF2-specific sgRNAs in the revised manuscript (Fig. S3G). Finally, we described the track example (Fig. 3I), changed the typo, and relabeled the panels.

Fig. S3H

REVIEWER COMMENTS

Reviewer #1 (Remarks to the Author):

In the revised manuscript, the authors address a number of issues with the original submission. In particular, I appreciate the new data and analyses of the PTEN gene and the occupancy of POU2AF2 near this gene. However, the authors have not remedied the main, underlying concern I have with the work. The title and sections of the text still claim that POU2AF2 carries out transcriptional repression, which is not supported by the data.

The additional experiments further support a model in which POU2AF2, like many cell type master regulators, interacts with SWI/SNF to open chromatin at regulatory elements. When POU2AF2 is depleted, SWI/SNF dissociates broadly from chromatin (from all three clusters of regulatory loci). This causes a number of regulatory sites to exhibit reduced accessibility. Over time, this causes a redistribution of Polycomb complexes and a dilution of H3K27me3 at some regulatory sites (called cluster 3 sites in this manuscript), which allows for reactivation of previously repressed loci. This phenomenon has been shown previously for OCT4 (POU5F1) and SWI/SNF in mouse embryonic stem cells by Rob Klose, Dirk Schubeler, Gerry Crabtree, etc. as the authors note in their response to reviewers.

The authors now seem to appreciate that the effects observed during their long term depletion are indirect – and that POU2AF2 is not directly acting as a repressor, but they have failed to update the title and text accordingly. If the authors could correct this, I would in principle be in favor of publishing the work. But as it stands, I fear that this manuscript will add confusion to the field.

Specific points:

1) the upregulation of genes located near cluster 3 loci (called here 'poised enhancers' although I see no evidence that they are in fact enhancers?) reaches statistical significance ($p=0.0012$) but is still very small. The authors should note in the text what the median fold upregulation is so that a reader can evaluate the magnitude of this effect.

2) I suggest that the authors not over-emphasize the clinical potential of the work. SWI/SNF inhibitors have not been promising in clinical testing and display extensive on-target toxicity. As shown in this work SWI/SNF inhibition causes effects that extend well beyond PTEN and cluster 3 associated genes. As recommended by the other reviewer, EZH2 inhibitors likely show greater promise. The authors are encouraged to edit their language (including in the abstract) about the clinical implications of this work so as to not be misleading.

Reviewer #3 (Remarks to the Author):

The authors have addressed the comments and made a significant improvement to the manuscript. It is publishable pending providing details of the CRISPR screen (Fig. 5A) in Material and Method section.

Reviewer #1 (Remarks to the Author):

In the revised manuscript, the authors address a number of issues with the original submission. In particular, I appreciate the new data and analyses of the PTEN gene and the occupancy of POU2AF2 near this gene. However, the authors have not remedied the main, underlying concern I have with the work. The title and sections of the text still claim that POU2AF2 carries out transcriptional repression, which is not supported by the data.

We thank the reviewer for acknowledging our improvement of the revised manuscript. Following the reviewer's suggestion, we changed the title and the text section to describe our findings more accurately.

The additional experiments further support a model in which POU2AF2, like many cell type master regulators, interacts with SWI/SNF to open chromatin at regulatory elements. When POU2AF2 is depleted, SWI/SNF dissociates broadly from chromatin (from all three clusters of regulatory loci). This causes a number of regulatory sites to exhibit reduced accessibility. Over time, this causes a redistribution of Polycomb complexes and a dilution of H3K27me3 at some regulatory sites (called cluster 3 sites in this manuscript), which allows for reactivation of previously repressed loci. This phenomenon has been shown previously for OCT4 (POU5F1) and SWI/SNF in mouse embryonic stem cells by Rob Klose, Dirk Schubeler, Gerry Crabtree, etc. as the authors note in their response to reviewers. The authors now seem to appreciate that the effects observed during their long term depletion are indirect – and that POU2AF2 is not directly acting as a repressor, but they have failed to update the title and text accordingly. If the authors could correct this, I would in principle be in favor of publishing the work. But as it stands, I fear that this manuscript will add confusion to the field.

We agree with the reviewer and changed the title and the text section to describe our findings more accurately.

Specific points:

1) the upregulation of genes located near cluster 3 loci (called here 'poised enhancers' although I see no evidence that they are in fact enhancers?) reaches statistical significance ($p=0.0012$) but is still very small. The authors should note in the text what the median fold upregulation is so that a reader can evaluate the magnitude of this effect.

We thank the reviewer for pointing out this. We mentioned the upregulation value in the text of our manuscript, and we highlighted it for the reviewer's convenience.

2) I suggest that the authors not over-emphasize the clinical potential of the work. SWI/SNF inhibitors have not been promising in clinical testing and display extensive on-target toxicity. As shown in this work SWI/SNF inhibition causes effects that extend well beyond PTEN and cluster 3 associated genes. As recommended by the other reviewer, EZH2 inhibitors likely show greater promise. The authors are encouraged to edit their language (including in the abstract) about the clinical implications of this work so as to not be misleading.

We thank both reviewers for pointing out this. We edited our statement in both abstract and main text following the reviewer's suggestion.

Reviewer #3 (Remarks to the Author):

The authors have addressed the comments and made a significant improvement to the manuscript. It is publishable pending providing details of the CRISPR screen (Fig. 5A) in Material and Method section.

We thank the reviewer for acknowledging our improvement of the revised manuscript. Following the reviewer's suggestion, we provided the details of the CRISPR screen presented in Figure 5A in the Materials and Method section.

REVIEWERS' COMMENTS

Reviewer #1 (Remarks to the Author):

The authors have made suggested changes and I support publication of this work.

Reviewer #3 (Remarks to the Author):

The authors have fully addressed my concerns. I have no more questions.